# Conformal Tradeoffs: Operational Profiles Beyond Coverage

## Abstract

Conformal prediction gives exact finite-sample coverage guarantees under exchangeability, but deployed systems are judged by more than coverage alone. For a fixed calibrated rule reused over a finite operational window, stakeholders also care about deployment-facing quantities such as commitment frequency, deferral, and decisive error exposure. These are not determined by coverage: calibration choices with similar coverage can still induce materially different operational profiles.

We study this characterization gap in a scoped setting: binary split conformal prediction under exchangeability with a fixed deployed rule. We introduce the Small-Sample Beta Correction (SSBC) which gives finite-sample coverage semantics for the deployed rule: it inverts the Beta/Beta–Binomial law governing calibration-conditional coverage to map a user request $(\alpha^\star, \delta)$ to the least conservative calibration grid point with calibration-conditional PAC semantics for the realized deployed rule. Calibrate-and-Audit then fixes the rule by calibration and uses an independent audit split to estimate the induced region–class label table, a reusable summary from which deployment-facing Key Performance Indicators (KPIs) follow by projection. Under this design, fixed operational rates admit exact finite-sample Binomial inference, while Beta–Binomial envelopes serve as practical predictive summaries for future windows. The induced partition also exposes regime boundaries, Pareto-relevant tradeoffs, and inverse-pricing questions for fixed downstream conventions.

Simulations validate the SSBC semantics and compare audit-based summaries with leave-one-out planning proxies; molecular toxicity data provide an audit-based empirical example, and a solubility case study illustrates scenario planning once coverage semantics are fixed.

## 1 Introduction: deployment-facing conformal prediction

Conformal prediction provides exact finite-sample coverage guarantees under exchangeability, but deployment decisions are not made from coverage alone (Vovk et al., 2005; Shafer & Vovk, 2008; Angelopoulos & Bates, 2023). Once a classifier is calibrated and reused over a finite operational window, stakeholders also care about commitment, deferral, and decisive-error exposure. Those quantities affect throughput and risk, yet they are not determined by marginal coverage.

We study this gap for binary split conformal prediction with a fixed deployed rule. Calibration $\tau$ fixes thresholds on score space $\mathcal{X}$, the thresholds induce a finite region partition $R_\tau(X)$ and associated fixed labels $Y$, and the deployed system is observed through the joint process $(R_\tau(X), Y)$. Our central claim is that this region–class structure is the deployment object of interest: it explains why coverage-matched rules can still behave differently in practice, and it provides a reusable summary from which many operational KPIs follow by projection.

Our method has two layers. The Small-Sample Beta Correction (SSBC) maps a user request $(\alpha^\star, \delta)$ to the least conservative conformal grid point with the intended finite-sample coverage semantics for the realized deployed rule. Calibrate–and–Audit then freezes that rule and uses an independent audit split to estimate deployment-facing rates. Under this design, fixed KPIs admit exact Binomial inference, while Beta–Binomial envelopes provide practical planning summaries for future windows.

---

Generative AI was used to assist in the preparation and formatting of this document.

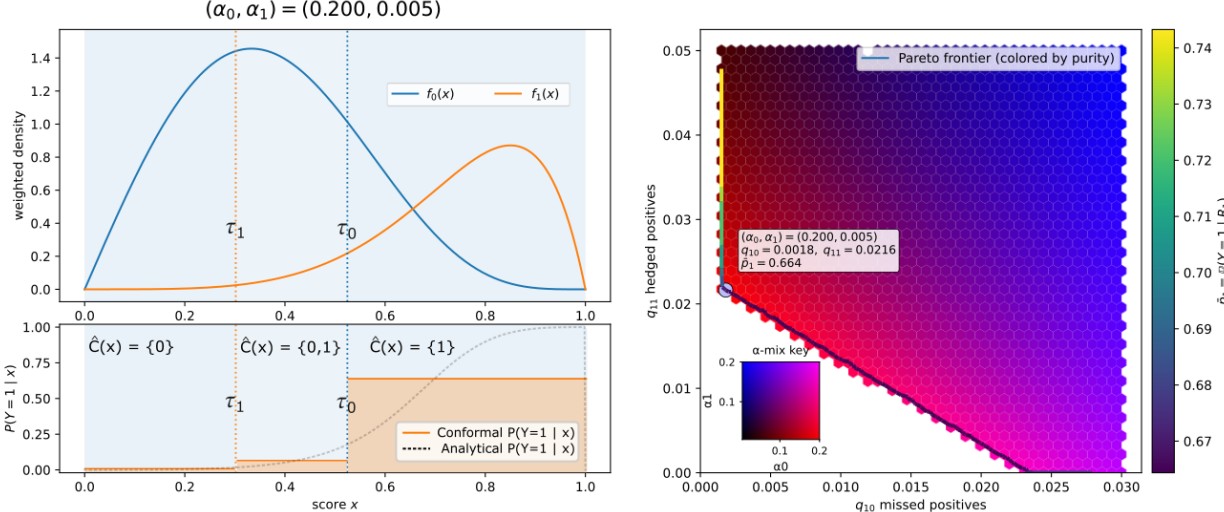

Figure 1: **Operational view of a calibrated conformal rule.** Calibration fixes a region partition; auditing the resulting region–class table yields a reusable operational summary; sweeping calibration settings traces feasible trade-offs.

The exact finite-sample guarantees in the paper are deliberately narrow: they apply to SSBC's coverage semantics for a fixed deployed rule and to audit-based inference for pre-declared KPIs at that same fixed rule. By contrast, calibration sweeps, Pareto views, leave-one-out proxies, and inverse-pricing analyses are planning tools rather than simultaneous certification statements.

## 1.1 Contributions

In the scoped setting of binary split conformal prediction under exchangeability, the paper makes three contributions.

**(1) Operational structure beyond coverage.** The calibration-induced partition constrains deployment behavior: changing thresholds reallocates mass across a small set of region types, so operational KPIs are coupled rather than independently tunable. This yields attainable sets, Pareto-relevant regimes, and a natural interface for asking which downstream action conventions are justified by the audited region-wise label composition.

**(2) SSBC as a coverage-semantics anchor.** SSBC inverts the exact rank/Beta law to map $(\alpha^\star, \delta)$ to the least conservative deployed grid point whose realized coverage satisfies a calibration-conditional PAC-style statement. It turns a nominal request into an explicit finite-sample semantic claim about the realized conformal rule.

**(3) Calibrate–and–Audit for deployment KPIs.** The region–class label table is the reusable audit object for a fixed rule. From it one can derive commitment, deferral, singleton-error, and related KPI rates by projection. Exact Binomial inference applies pointwise to fixed audit-based KPIs, while leave-one-out proxies are used only for exploratory planning when an audit split is unavailable.

## 1.2 Relation to prior work

Conformal prediction constructs set-valued predictors with finite-sample coverage guarantees under exchangeability (Vovk et al., 2005; Shafer & Vovk, 2008; Angelopoulos & Bates, 2023). Conformal risk control (CRC) extends this perspective by selecting thresholds to satisfy user-specified *scalar* risk constraints under related assumptions (Angelopoulos et al., 2024; Bates et al., 2021), and other work studies stronger

conditional notions of validity or connects conformal methods to downstream decision-making (Tibshirani et al., 2019; Fannjiang et al., 2022; Bian & Barber, 2023; Gibbs et al., 2025; Lekeufack et al., 2023; Kiyani et al., 2025). These studies typically ask how to guarantee coverage, a chosen scalar risk functional, or decision quality under a specified objective. Our emphasis is different: once a conformal rule is fixed, what operational behavior should one expect from it, and how much of that behavior is already determined by calibration geometry rather than by coverage alone?

Recent Human-Computer Interaction (HCI)-oriented work raises a related deployment concern: conformal sets can be a "murky" interface when validity summaries do not translate cleanly into action-relevant consequences (Hullman et al., 2025). Our region–class label table view puts that issue on a concrete footing: it audits how mass falls across region types and labels, exposing how often a rule commits, hedges, or abstains, and where decisive errors arise. This makes clear how coverage-matched rules can still behave differently in deployment. We do not replace validity guarantees; we clarify what they do and do not imply for deployment behavior. Once constructed, the table supports many KPIs and policy projections without re-running calibration.

Closest in spirit to our viewpoint is the inverse CRC formulation of Zhou & Zhu (2025), who trace certified miscoverage–regret trade-offs for predict-then-optimize pipelines, and related work on conformal efficiency optimizes set-size functionals subject to validity constraints (Yang & Kuchibhotla, 2021). Our objective is different: rather than introducing another scalar criterion, we make the *deployment interface* itself the primitive. A calibration choice $\theta$ fixes a finite conformal partition, and the induced region–class label table is the minimal reusable summary that determines *many* operational KPIs by linear projection under any fixed region-based policy; this yields an *operational profile* (rate-vector) map $\theta \mapsto \mathbf{r}(\theta)$ and an attainable set of behaviors that is geometrically constrained, so coverage-matched rules can still induce qualitatively different deployment regimes. Methodologically, we enforce a disciplined two-stage workflow: calibration sweeps and KPI estimates on the audit split, or LOO surrogates when an audit split is unavailable, are used only to expose feasible regimes and trade-offs, while exact finite-sample guarantees are asserted *pointwise* for pre-declared KPIs at a fixed deployed rule using an independent audit split via Binomial inference (with Beta–Binomial envelopes as planning summaries for finite windows). SSBC plays a complementary role as a *semantics anchor*, mapping a user request $(\alpha^\star, \delta)$ to a concrete calibration grid point so the operational map is navigated relative to an explicit finite-sample coverage semantics for the realized rule.

### 1.3 Roadmap

The remainder of the paper is organized as follows: Section 2 defines the fixed deployed object and the region–class label table, Section 3 presents SSBC and Calibrate–and–Audit, Section 4 gives the binary geometric interpretation, Section 5 reports simulation, molecular toxicology, and solubility scenario planning, and Section 6 concludes. Technical derivations and extended examples are deferred to the appendices.

## 2 Setting and notation: calibration-conditional viewpoint

We study a *single deployed classifier*: a scoring model is trained once, treated as fixed, and then calibrated once by split conformal prediction. Randomness therefore comes from the calibration draw and future deployment examples, not from retraining or repeated recalibration. To evaluate deployment-facing quantities beyond coverage, we reserve an exchangeable audit split that is never used to choose thresholds.

### 2.1 Data splits and exchangeability assumptions

Let $\mathcal{D}_{\mathrm{train}}$ denote the training data used to fit a base score model. After training, the score function is treated as fixed. Let

$$\mathcal{D}_{\mathrm{cal}} = \{(X_i^c, Y_i^c)\}_{i=1}^{n_{\mathrm{cal}}}, \qquad \mathcal{D}_{\mathrm{audit}} = \{(X_i^a, Y_i^a)\}_{i=1}^{n_{\mathrm{audit}}}.$$

The calibration split $\mathcal{D}_{\mathrm{cal}}$ is used once to set the deployed thresholds, while $\mathcal{D}_{\mathrm{audit}}$ is reserved for post-calibration evaluation of fixed-rule KPIs.

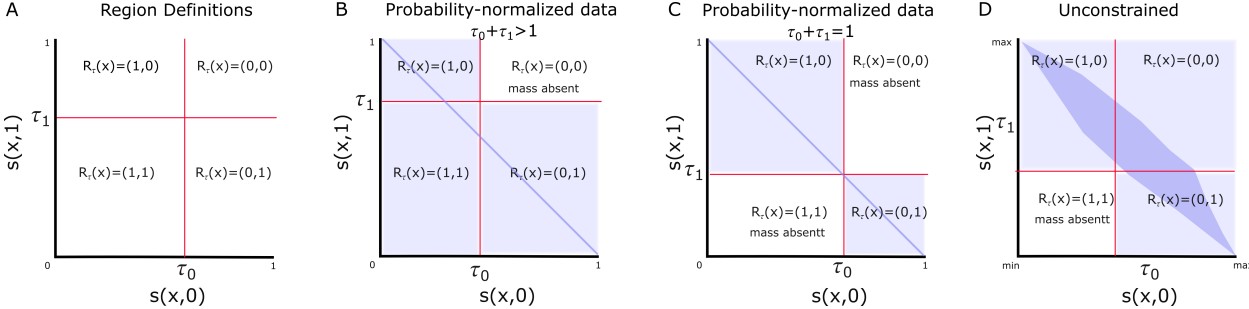

Figure 2: **Thresholds induce a finite region partition.** In the binary probability-normalized setting, the score support determines which regions can carry mass under a fixed pair of thresholds.

We assume calibration, audit, and future deployment points are jointly exchangeable conditional on the fixed trained score model. We focus on binary classification because it exposes the geometry and deployment trade-offs most cleanly.

## 2.2 Scores and calibration thresholds

Let $\mathcal{Y} = \{1, \dots, K\}$, and let $s : \mathcal{X} \times \mathcal{Y} \to \mathbb{R}$ be a nonconformity score (Shafer & Vovk, 2008; Lei et al., 2018). In the experiments we use $s(x, y) = 1 - P(y \mid x)$, but the setup below does not depend on that choice.

Given $\mathcal{D}_{\mathrm{cal}}$, compute calibration scores

$$S_i := s(X_i^c, Y_i^c), \qquad i = 1, \dots, n_{\mathrm{cal}},$$

and let $S_{(1)} \leq \cdots \leq S_{(n_{\mathrm{cal}})}$ denote their order statistics. Split conformal calibration selects

$$\tau := S_{(k)},$$

where $k \in \{1, \dots, n_{\mathrm{cal}}\}$ is determined by the requested miscoverage level. Equivalently, one may index the same grid by $u := n_{\mathrm{cal}} + 1 - k$, so the deployed grid level is $\alpha_{\mathrm{grid}} = u/(n_{\mathrm{cal}} + 1)$.

We use class-conditional split conformal (Vovk, 2012a;b), so thresholds $\tau_y$ are computed separately within each class. For a fixed calibration draw, this yields a threshold vector $\tau = (\tau_1, \dots, \tau_K)$ that remains fixed during deployment.

## 2.3 Regions, observability, and policies

Once thresholds are fixed, they induce a finite region label. Writing

$$s(x) := (s(x, 1), \dots, s(x, K)), \qquad \tau := (\tau_1, \dots, \tau_K),$$

the region map is

$$R_\tau(x) := \big( \mathbf{1}\{s(x, 1) \leq \tau_1\}, \dots, \mathbf{1}\{s(x, K) \leq \tau_K\} \big) \in \{0, 1\}^K.$$

Thus calibration fixes a finite partition of score space. The partition itself depends only on $\tau$, while the amount of mass carried by each region depends on the deployment distribution. This partition is the paper's operational interface (Figure 2).

Deployment is observed through the joint process $(R_\tau(X), Y)$. To make downstream actions explicit, we also allow a fixed deployment policy $\pi : \mathcal{R} \to 2^{\mathcal{Y}}$ and write

$$\hat{C}_\pi(x) := \pi(R_\tau(x)).$$

The standard conformal predictor is the set-inclusion policy

$$\pi_{\mathrm{SI}}(r) := \{y \in \mathcal{Y} : r_y = 1\}, \qquad \hat{C}_{\pi_{\mathrm{SI}}}(x) = \{y \in \mathcal{Y} : s(x, y) \leq \tau_y\}.$$

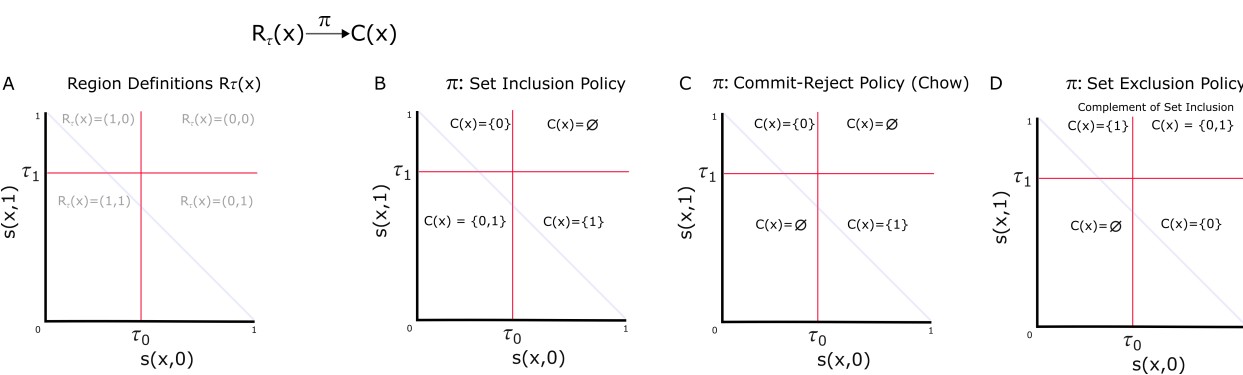

Figure 3: **Deployment policies as projections on a fixed region structure.** Different region-based policies act on the same calibrated partition but induce different reported outputs.

As an example beyond set inclusion, consider a region-triggered commit policy: in the binary case, choose a trigger set $A \subseteq \{0,1\}^2$ and an action $a \in \{0,1\}$, then commit to $\{a\}$ when $R_\tau(x) \in A$ and abstain otherwise. This simple template makes the calibration/action separation concrete: calibration fixes the region partition, while policy chooses how to act on the realized region. Additional binary projection masks can be found in Appendix G.

### 2.4 Auditing primitives: region–class label tables and policy projections

Conditional on $\mathcal{D}_{\mathrm{cal}}$, the deployed rule is fixed, and the primitive observable for auditing is the pair $(R_{\tau(\theta)}(X), Y)$ on an exchangeable labeled split. Here $\theta \in \Theta$ indexes thresholds $\tau(\theta)$ and hence the region map $R_{\tau(\theta)}$. Two linked objects will be used throughout the paper.

**(A) Region–class label joint table.** Define the calibration-conditional joint probabilities

$$p_{r,y}(\theta) := \mathbb{P}\big(R_{\tau(\theta)}(X) = r, \ Y = y \mid \mathcal{D}_{\mathrm{cal}}\big), \qquad (r,y) \in \mathcal{R} \times \mathcal{Y}.$$

These are fixed but unknown constants for the deployed rule. The audit split estimates them through

$$K_{r,y}^{\mathrm{audit}}(\theta) := \sum_{i=1}^{n_{\mathrm{audit}}} \mathbf{1}\{R_{\tau(\theta)}(X_i^a) = r, \ Y_i^a = y\}, \qquad \widehat{p}_{r,y}^{\,\mathrm{audit}}(\theta) = \frac{K_{r,y}^{\mathrm{audit}}(\theta)}{n_{\mathrm{audit}}}.$$

Because the cells $(r,y)$ form a finite partition, the table sums to one and changing $\theta$ reallocates mass across regions. Those conservation constraints drive the geometric coupling studied later.

**(B) Policy-specific indicators.** Fix a deployment policy $\pi$. Many KPIs of interest can be written as a Bernoulli indicator

$$I_\ell := g_\ell(R_{\tau(\theta)}(X), Y; \pi) \in \{0,1\}, \qquad p_\ell(\theta) := \mathbb{P}\big(I_\ell = 1 \mid \mathcal{D}_{\mathrm{cal}}\big).$$

For example, $g_{\mathrm{abs}}(r,y;\pi) = \mathbf{1}\{\pi(r) = \varnothing\}$ and $g_{\mathrm{err}}(r,y;\pi) = \mathbf{1}\{|\pi(r)| = 1, \ y \notin \pi(r)\}$. The corresponding audit count is

$$K_\ell^{\mathrm{audit}}(\theta) := \sum_{i=1}^{n_{\mathrm{audit}}} g_\ell(R_{\tau(\theta)}(X_i^a), Y_i^a; \pi), \qquad \widehat{p}_\ell^{\,\mathrm{audit}}(\theta) = \frac{K_\ell^{\mathrm{audit}}(\theta)}{n_{\mathrm{audit}}}.$$

Thus the region–class label table is the reusable audit object, and KPI rates are its projections.

**Projection identity.** The table and KPI views are linked by linearity:

$$K_\ell^{\mathrm{audit}}(\theta) = \sum_{r \in \mathcal{R}} \sum_{y \in \mathcal{Y}} g_\ell(r,y;\pi) \, K_{r,y}^{\mathrm{audit}}(\theta), \qquad \widehat{p}_\ell^{\,\mathrm{audit}}(\theta) = \frac{K_\ell^{\mathrm{audit}}(\theta)}{n_{\mathrm{audit}}}. \tag{1}$$

Taking expectation gives

$$p_\ell(\theta) = \sum_{r \in \mathcal{R}} \sum_{y \in \mathcal{Y}} g_\ell(r, y; \pi) \, p_{r,y}(\theta).$$

In words: estimate the table once, then obtain policy-level KPIs by projection.

**Worked example: coverage as a projection of the region–class label table.** Under set inclusion, coverage is the sum of the region–class label cells where the true label lies in the predicted set. Consider $\mathcal{Y} = \{0, 1\}$ and the set-inclusion policy $\pi_{\mathrm{SI}}$. For fixed thresholds, each input falls into one of four regions $\mathcal{R} = \{r_{10}, r_{11}, r_{01}, r_{00}\}$, and the primitive auditing object is $(R_\tau(X), Y) \in \mathcal{R} \times \mathcal{Y}$. The corresponding region–class label table is

$$P(\theta): \quad \begin{array}{c|ccc} & y = 0 & y = 1 & \hat{C}_{\pi_{\mathrm{SI}}}(X) \\ \hline r_{10} = (1,0) & \mathbf{p_{10,0}}(\theta) & p_{10,1}(\theta) & \{0\} \\ r_{11} = (1,1) & \mathbf{p_{11,0}}(\theta) & \mathbf{p_{11,1}}(\theta) & \{0,1\} \\ r_{01} = (0,1) & p_{01,0}(\theta) & \mathbf{p_{01,1}}(\theta) & \{1\} \\ r_{00} = (0,0) & p_{00,0}(\theta) & p_{00,1}(\theta) & \varnothing \end{array} \qquad \text{with} \qquad \sum_{r \in \mathcal{R}} \sum_{y \in \mathcal{Y}} p_{r,y}(\theta) = 1.$$

Under $\pi_{\mathrm{SI}}$, the event $\{Y \in \hat{C}_{\pi_{\mathrm{SI}}}(X)\}$ corresponds to the four bold cells $(r_{10}, 0)$, $(r_{11}, 0)$, $(r_{11}, 1)$, and $(r_{01}, 1)$, hence

$$p_{\mathrm{cov}}(\theta) = p_{10,0}(\theta) + p_{11,0}(\theta) + p_{11,1}(\theta) + p_{01,1}(\theta) = 1 - \big(p_{10,1}(\theta) + p_{01,0}(\theta) + p_{00,0}(\theta) + p_{00,1}(\theta)\big).$$

The calibration choices $(\alpha_0^\star, \alpha_1^\star)$ determine the class-conditional miscoverage primitives $p_{10,1}(\theta) + p_{00,1}(\theta)$ for $Y = 1$ and $p_{01,0}(\theta) + p_{00,0}(\theta)$ for $Y = 0$, while the decomposition of these totals into wrong-singleton errors versus abstentions is dictated by the deployment distribution. This makes explicit that (i) coverage is a projection, (ii) once thresholds are fixed it is determined by how mass is redistributed across regions, and (iii) deployment-facing consequences are not determined by the calibration targets alone.

The same "sum selected cells" structure applies to abstention/deferral, decisiveness, and decisive error exposure under any fixed policy. Appendix G records additional binary projection masks and related book-keeping.

## 3 Operational quantities as first-class objects

Deployment behavior is summarized by operational rates induced by a calibrated partition together with a fixed deployment policy $\pi$. The purpose of this section is to separate what is certifiable for a fixed deployed rule from what is useful only for planning across candidate rules. We first fix the coverage semantics of the deployed rule through SSBC, then turn to audit-based inference and comparative planning at that fixed rule. We keep three layers distinct: geometry (the induced regions), policy (the projection from regions to outputs), and rates (the auditable deployment frequencies).

### 3.1 Calibrate–and–Audit

Beyond marginal coverage, finite-sample evaluation of operational rates requires an independent audit split. If $\mathcal{D}_{\mathrm{cal}}$ is reused both to choose thresholds and to evaluate downstream indicators, the resulting event counts are no longer Bernoulli draws conditional on a fixed rule. Calibrate–and–Audit avoids this coupling by freezing thresholds on $\mathcal{D}_{\mathrm{cal}}$ and evaluating operational quantities on $\mathcal{D}_{\mathrm{audit}}$ only.

When an audit split is unavailable, we use leave-one-out (LOO) recalibration as a practical proxy. Throughout the paper, that proxy is used only for exploration and scenario planning, not for the exact fixed-rule guarantees. The structural issue is that calibration reuse makes the threshold random with respect to the same observations used for evaluation, inducing dependence between selection and KPI counts rather than conditionally i.i.d. Bernoulli trials. Appendix D gives the covariance argument and the details of the LOO construction.

## 3.2 SSBC as a calibration navigation coordinate

We use coverage as the semantic anchor for navigating the calibration grid. Split conformal calibration lives on a finite grid indexed by $u \in \{1, \ldots, n_{\mathrm{cal}}\}$ with $\alpha_{\mathrm{grid}} = u/(n_{\mathrm{cal}} + 1)$; see Section 2.2. SSBC maps $(\alpha^\star, \delta)$ to the least conservative admissible index $u^\star(\alpha^\star, \delta)$ satisfying

$$\mathbb{P}_{\mathcal{D}_{\mathrm{cal}}}\Big(\mathbb{P}\big(Y \in \hat{C}(X) \mid \mathcal{D}_{\mathrm{cal}}\big) \geq 1 - \alpha^\star\Big) \geq 1 - \delta.$$

Thus SSBC turns a semantic request into a concrete deployed threshold choice. In the binary class-conditional setting this yields

$$(u_0^\star, u_1^\star) = \big(u^\star(\alpha_0^\star, \delta_0),\ u^\star(\alpha_1^\star, \delta_1)\big),$$

which determine $(\tau_0, \tau_1)$ and therefore the calibration setting $\theta$. Although the user-facing request is four-dimensional, $(\alpha_0^\star, \delta_0, \alpha_1^\star, \delta_1)$, calibration collapses it to a low-dimensional navigation coordinate on the deployed grid. The later planning objects are therefore indexed by a coverage-semantically meaningful choice of deployed rule rather than by an abstract threshold sweep.

## 3.3 Audit-based predictive envelopes for future windows

Under Calibrate–and–Audit, conditional on $\mathcal{D}_{\mathrm{cal}}$ the deployed rule is fixed and audit set is exchangeable with future deployment points. Therefore, for any fixed KPI indicator $I_\ell$,

$$K_\ell^{\mathrm{audit}}(\theta) \mid \mathcal{D}_{\mathrm{cal}} \sim \mathrm{Binomial}\big(n_{\mathrm{audit}}, p_\ell(\theta)\big),$$

and for a future window of size $m$,

$$K_\ell^m(\theta) \mid \mathcal{D}_{\mathrm{cal}} \sim \mathrm{Binomial}\big(m, p_\ell(\theta)\big).$$

These Binomial laws support exact fixed-rule inference for the latent rate $p_\ell(\theta)$; for example, Clopper–Pearson intervals (Clopper & Pearson, 1934) are valid from the audit count $K_\ell^{\mathrm{audit}}(\theta)$. For finite future windows we use Beta–Binomial envelopes as practical planning summaries (Skellam, 1948; Johnson et al., 1997; Gelman et al., 2013).

## 3.4 Rate vectors, attainable operational sets, and Pareto filtering

The guarantees above are *pointwise*: they certify a pre-declared KPI at a fixed deployed rule, not an adaptively selected member of a sweep. Planning is comparative. For a calibration family $\Theta$, a fixed policy $\pi$, and a KPI list, each setting $\theta \in \Theta$ determines an operational profile

$$\mathbf{p}(\theta) = \big(p_1(\theta), \ldots, p_L(\theta)\big),$$

whose coordinates are obtained by projection from the same region–class label table via (1). Sweeping $\theta$ traces the attainable set

$$\mathcal{V}(\Theta; \pi) := \{\mathbf{p}(\theta) : \theta \in \Theta\} \subset \mathbb{R}^L.$$

Because the underlying table obeys conservation constraints, changing $\theta$ reallocates mass across regions rather than tuning rates independently.

To summarize planning-relevant regimes without committing to a scalar objective, we use an oriented Pareto filter. After declaring which KPIs are preferable to increase or decrease, a setting is *nondominated* if no other setting is at least as good in every oriented coordinate and strictly better in one. These fronts are planning objects; exact finite-sample guarantees remain pointwise for fixed settings and fixed KPIs. Section 4 now makes precise why these attainable sets are geometrically constrained and why sweeping SSBC-indexed settings produces coupled, regime-dependent trade-offs. Any later scalar objective that is monotone in the chosen orientation must attain its optimum on this front (Miettinen, 1999). The follow-on question is inverse pricing: which downstream cost ratios justify a fixed action convention at a Pareto-relevant regime? Further derivations are deferred to Appendices C and I.10.

# 4 Consequences of a fixed conformal partition in the binary case

This section isolates the main binary geometric fact behind the operational trade-offs. Once thresholds are fixed, deployment behavior is governed by the joint region–class table $(R_\tau(X), Y)$. Under probability-normalized scores, the threshold pair $(\tau_0, \tau_1)$ cannot move operational KPIs independently; it reallocates mass across a small number of region types. Equivalently, this section makes precise the conservation-constraint intuition behind the attainable sets and Pareto filtering introduced in Section 3.4. In the binary case this yields three practical implications: coupled feasibility, region-wise policy dependence, and decision optimality determined by the audited table rather than by coverage alone. Appendix E contains the full construction and proofs, while Appendix F develops the decision-theoretic consequences.

## 4.1 Binary conformal geometry and regime boundaries

Let $\mathcal{Y} = \{0, 1\}$ and fix class-conditional thresholds $\tau = (\tau_0, \tau_1)$. Once calibrated, the object that governs operational characteristics in deployment is the region label

$$R_\tau(x) := \left(\mathbf{1}\{s(x, 0) \le \tau_0\}, \ \mathbf{1}\{s(x, 1) \le \tau_1\}\right) \in \{0, 1\}^2,$$

with outcomes $10, 11, 01, 00$, which under the set inclusion policy $\pi_{\mathrm{SI}}$ correspond to singleton-0, hedge, singleton-1, and abstention, respectively.

**Probability-normalized scores induce a regime boundary.** For probability-normalized scores $s(x, y) = 1 - P(y \mid x)$ we have $s(x, 0) + s(x, 1) = 1$. Hence a point cannot satisfy both class thresholds unless $\tau_0 + \tau_1 \ge 1$, and it cannot violate both unless $\tau_0 + \tau_1 \le 1$. Consequently:

- **Hedging regime $(\tau_0 + \tau_1 > 1)$:** 11 may occur and 00 cannot (singletons + hedges; no abstention).

- **Rejection regime $(\tau_0 + \tau_1 < 1)$:** 00 may occur and 11 cannot (singletons + abstention; no hedges).

- **Boundary $(\tau_0 + \tau_1 = 1)$:** only 10 and 01 occur; under $\pi_{\mathrm{SI}}$ outputs are always singletons.

Crossing $\tau_0 + \tau_1 = 1$ therefore changes which outcome types are even feasible. This deterministic boundary is the simplest geometric reason that changing thresholds can produce qualitatively different operational regimes.

**Cross-threshold coupling within regimes.** Within the hedging regime $(\tau_0 + \tau_1 > 1)$, the diagonal support is partitioned into three contiguous intervals. Parameterize by $u = s(x, 0)$ (so $s(x, 1) = 1 - u$):

$$u \in [0, 1 - \tau_1) \ \Rightarrow \ R_\tau(x) = 10, \qquad u \in [1 - \tau_1, \tau_0] \ \Rightarrow \ R_\tau(x) = 11, \qquad u \in (\tau_0, 1] \ \Rightarrow \ R_\tau(x) = 01.$$

Thus region boundaries are governed by *opposing* thresholds: changing $\tau_1$ moves the boundary controlling singleton-0 mass, while changing $\tau_0$ moves the boundary controlling singleton-1 mass. Thresholds are therefore mass-reallocation boundaries, not independent class-wise knobs.

For planning, the key consequence is that sweeping calibration settings can cross regime boundaries and induce discontinuous changes in which outputs are even feasible. This is the geometric source of the coupled attainable sets and Pareto fronts seen later in the experiments.

## 4.2 Region observability and interface-relative decision optimality

Fix $\tau$ and treat the region label $R_\tau(X) \in \{0, 1\}^2$ as the deployed observable. The joint region–class label probabilities

$$p_{r,y} := \mathbb{P}\left(R_\tau(X) = r, \ Y = y \mid \mathcal{D}_{\mathrm{cal}}\right), \qquad r \in \{0, 1\}^2, \ y \in \{0, 1\},$$

fully characterize the information available to any downstream rule that uses only this interface. Write $p_r := p_{r,0} + p_{r,1}$ for region mass and define the within-region label frequency

$$\eta_r := \Pr(Y = 1 \mid R_\tau(X) = r, \ \mathcal{D}_{\mathrm{cal}}) = \frac{p_{r,1}}{p_r} \quad (p_r > 0).$$

Because the interface takes finitely many values, both evaluation and decision-making reduce to functions of the same table $\{p_{r,y}\}$ (equivalently $\{p_r, \eta_r\}$). In particular, a region-based action convention is justified by the *within-region* label composition, not by marginal coverage alone and not by the set-valued output alone.

Formally, let $\mathcal{A}$ be an action set and let $L(a, y)$ denote the loss of taking action $a \in \mathcal{A}$ when $Y = y$. A region-based policy $\tilde{\pi} : \{0, 1\}^2 \to \mathcal{A}$ is *decision-optimal relative to the conformal interface* if, for every populated region $r$,

$$\tilde{\pi}(r) \in \arg\min_{a \in \mathcal{A}} \mathbb{E}[L(a, Y) \mid R_\tau(X) = r, \ \mathcal{D}_{\text{cal}}].$$

In words, once the deployed interface reveals only the region label $r$, the optimal action is the one with smallest expected loss under the label mix within that region. This optimality is determined region-wise by the within-region label frequency $\eta_r$.

For the main text, the key implication is simply that the same audited region–class table used for operational planning also determines whether a fixed region-based convention is rational under a stated cost model. The full Chow-style inverse-pricing analysis is deferred to Appendix F, but two qualitative consequences are worth flagging here: the resulting inverse-pricing envelope is polyhedral in cost-ratio coordinates, and coverage-matched settings can induce different, even disjoint, pricing envelopes because the region composition changes.

## 5 Results: evidence for coverage semantics, audit behavior, and planning

This section keeps the empirical story at the same scope as the theory. We validate SSBC's coverage semantics in simulation, use a toxicology dataset as an audit-based example of fixed-rule operational evaluation, and close with a solubility scenario-planning illustration in which the model is fixed and only the calibration layer is varied.

### 5.1 Numerical simulations: coverage semantics first, operational summaries second

### 5.1.1 Coverage: numerical realization of SSBC guarantees

We first isolate the finite-sample law underlying SSBC. Calibration nonconformity scores are drawn i.i.d. from a continuous heavy-tailed reference distribution, and we compare nominal split conformal, a one-sided DKWM correction (Massart, 1990; Dvoretzky et al., 1956; Vovk, 2012a), and SSBC at $(\alpha^\star, \delta) = (0.10, 0.10)$ over a finite deployment window of size $m_{\text{infer}} = 100$.

Nominal split conformal under-controls calibration-conditional risk in this finite-window view, whereas DKWM enforces the target through strong conservatism. Across representative calibration sizes, nominal split conformal produces violation probabilities far above the requested level, DKWM drives the same probabilities well below target by substantial conservatism, and SSBC tracks the intended finite-window semantics much more closely up to unavoidable grid effects. For example, at $n_{\text{cal}} = 100$ the observed coverage violation rate is 0.4075 for nominal split conformal, 0.0004 for DKWM, and 0.0960 for SSBC, close to the target $\delta = 0.10$. The full calibration-size grid with theoretical and observed violation rates is reported in Appendix C, Table 3.

### 5.1.2 Operational rate summaries: LOO versus two-sample audit reference

We next study operational quantities beyond coverage. The goal is to compare the two-sample Calibrate–and–Audit constructed Beta–Binomial predictive summaries to a single-sample LOO surrogate intended for feasibility exploration.

**Synthetic probability model.** We use a controlled binary probability model. Each draw produces $Y \in \{0, 1\}$ with $\mathbb{P}(Y = 1) = p_{\text{class}}$ for $p_{\text{class}} \in \{0.10, 0.50\}$, and

$$P_1 \mid Y = 1 \sim \text{Beta}(a, b), \qquad P_1 \mid Y = 0 \sim \text{Beta}(2, 7),$$

with $(a, b) \in \{(4, 3), (9, 3)\}$. We then set $(P_0, P_1) = (1 - P_1, P_1)$ and use class-conditional scores $S_y := 1 - P_y$.

**Two-sample reference and LOO surrogate.** For each configuration, we draw independent datasets $\mathcal{D}_1$ and $\mathcal{D}_2$ of size $N = 500$, calibrate on $\mathcal{D}_1$ with SSBC-adjusted thresholds at $(\alpha, \delta) = (0.10, 0.10)$, freeze the rule, and evaluate operational indicators on $\mathcal{D}_2$. This yields the two-sample audit reference. As a single-sample surrogate, we recompute thresholds leave-one-out, pool the resulting indicators, and map them to planning envelopes, optionally widened by controlled pessimism (Appendix D).

Figure 4 shows that, in these simulated geometries, LOO-based envelopes align closely with the two-sample reference, while inflation widens intervals without shifting centers. This supports LOO as a practical planning proxy when an explicit audit split is unavailable, while keeping the main operational evidence tied to the two-sample design.

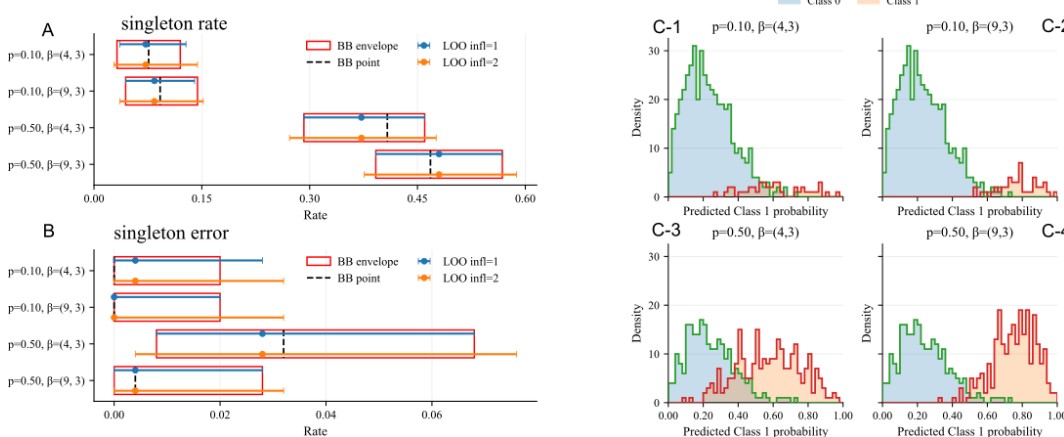

Figure 4: Operational rate envelopes and score geometry. **(A)** Singleton rate and **(B)** singleton error for two class prevalences and two class 1 generating distributions, with class 0 drawn from Beta$(2,7)$. Red rectangles denote the two-sample Beta–Binomial future-window summary and the dashed vertical line its center; blue and orange intervals show leave-one-out envelopes under two inflation levels. **(C1–C4)** Histograms of predicted class 1 probability by true class make explicit how score geometry shapes singleton mass, singleton error, and the asymmetry of the envelopes.

### 5.2 Tox21: an empirical example of the audit-based operational view

Tox21 (Mayr et al., 2016; Huang et al., 2016) stress-tests the framework under severe class imbalance, where class-conditional calibration counts can be small. Across twelve endpoints and 100 random train/calibration/audit splits, we compare nominal split conformal, DKWM, and SSBC at $(\alpha, \delta) = (0.10, 0.10)$. Dataset composition and representative endpoint-level operational tables are deferred to Appendix H.

The aggregate calibration-conditional picture follows the same trend as the numerical simulation: nominal split conformal again shows elevated coverage violation in this conditional small-$n$ regime, DKWM suppresses violations through strong conservatism, and SSBC lands much closer to the intended finite-sample semantics while avoiding DKWM's excess inflation. The set-size trade-off is similarly clear: mean set size is 1.41 for nominal split conformal, 1.78 for DKWM, and 1.54 for SSBC, with singleton frequency ordered in the opposite direction. The aggregated coverage and set-size summary is reported in Appendix H, Table 5.

#### 5.2.1 Operational summaries on an independent evaluation split

With coverage semantics fixed by SSBC at $(\alpha, \delta) = (0.10, 0.10)$, we examine the induced region–class summaries on an independent audit split. The point of the endpoint-level table is different from the aggregate coverage table above: it shows what the fixed rule actually does in deployment-facing KPI terms and how closely the LOO proxy tracks that held-out operational picture.

Table 1: **Representative Tox21 endpoint: SR-MMP.** Joint rates are normalized by the endpoint test-set size. The $\mathcal{D}_1$ LOO columns provide planning summaries (Point Estimate PE and 95% Prediction Interval PI) from calibration data, while the $\mathcal{D}_2$ columns report the independent audit reference and its predictive interval.

| Operational quantity | Class | LOO PE $\mathcal{D}_1$ | LOO 95% PI $\mathcal{D}_1$ | PE $\mathcal{D}_2$ | BB 95% PI $\mathcal{D}_2$ |
|---|---|---|---|---|---|
| Singleton rate | Class 0 | 0.662 | $[0.604, 0.718]$ | 0.651 | $[0.615, 0.686]$ |
| | Class 1 | 0.130 | $[0.092, 0.173]$ | 0.117 | $[0.094, 0.142]$ |
| Doublet rate | Class 0 | 0.163 | $[0.121, 0.211]$ | 0.201 | $[0.172, 0.231]$ |
| | Class 1 | 0.045 | $[0.023, 0.074]$ | 0.031 | $[0.020, 0.046]$ |
| Wrong-singleton rate | Class 0 | 0.073 | $[0.044, 0.107]$ | 0.070 | $[0.052, 0.090]$ |
| | Class 1 | 0.012 | $[0.002, 0.030]$ | 0.011 | $[0.005, 0.020]$ |

For SR-MMP, the LOO proxy and the independent audit split agree closely on the main operational picture: most mass lies in singleton predictions for class 0, class 1 singleton mass is smaller but still stable, and wrong-singleton rates remain low relative to total singleton mass. This is the intended role of the endpoint table in the paper: not to certify the LOO proxy, but to show that the audit-based operational view yields a compact, interpretable deployment summary at a fixed rule. A second endpoint illustrating a lower-prevalence regime is kept in Appendix H.

### 5.3 Solubility: scenario planning once coverage is fixed

The solubility case study is a planning illustration of the attainable operational KPI trade-offs. We train a fixed model on AquaSolDB (Sorkun et al., 2019), restrict calibration to a lipophilic deployment scenario, and use the LOO planning interface on the scenario-restricted sample to expose the feasible operating regimes once coverage semantics have been fixed. Figure 5 summarizes the resulting planning interface. The left panel shows that sweeping SSBC settings does not induce a simple monotone trade-off between soluble-class exclusion, deferral, and decisive-correct mass; instead it yields a constrained attainable set with a small set of Pareto-relevant regimes. The right panel shows the associated inverse-pricing screen for a fixed downstream convention, illustrating that the same Pareto-relevant regimes need not remain decision-optimal under the same cost-ratio assumptions.

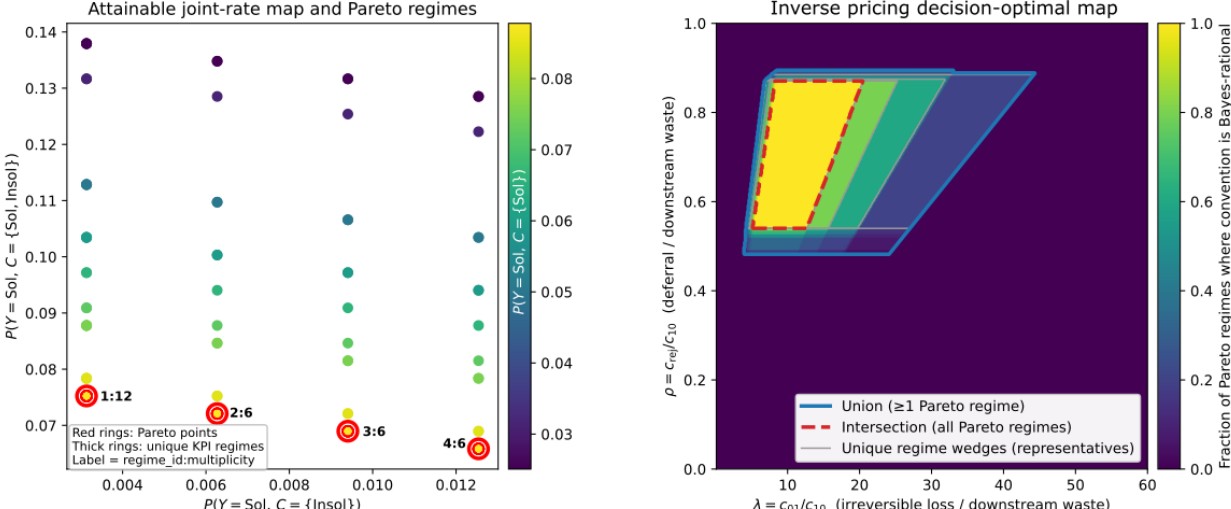

Figure 5: **Solubility planning and inverse pricing.** Left: attainable planning regimes induced by sweeping SSBC settings on a restricted deployment scenario. Right: cost-ratio regions for which a fixed downstream convention is decision-optimal relative to the conformal interface.

The main message is qualitative: once coverage semantics are fixed, the same conformal interface can support multiple planning regimes with materially different operational profiles. Detailed KPI tables for selected Pareto-optimal regimes, scenario construction, and parameter diagnostics are listed in Appendix I.

## 6 Discussion & Conclusions

For a fixed deployed binary split-conformal rule under exchangeability, coverage and deployment behavior are not the same object. SSBC gives a finite-sample semantic interpretation of the coverage request, and Calibrate–and–Audit evaluates the resulting fixed rule through the induced region–class table. That table is the reusable deployment summary in this setup: it supports KPI estimation, planning views, and downstream decision-theoretic questions without retraining the base model.

This is the paper's intended division of labor. SSBC answers what coverage claim is being made about the realized deployed rule; Calibrate–and–Audit answers what that same fixed rule is likely to do operationally over a finite window. The geometric analysis explains why both layers are needed: thresholds reallocate mass across a fixed partition, so operational KPIs are coupled.

The simulations and case studies should be read at exactly that scope: the simulations validate the intended SSBC semantics under calibration randomness, the Tox21 study shows that an independent audit split yields interpretable fixed-rule KPI summaries in a realistic small-sample regime, and the AquaSolDB example shows how the same interface can be used for planning once coverage semantics are fixed. Taken together, these experiments support a division of labor that we regard as practically useful: use SSBC to decide what coverage claim is being made about the deployed rule, then use an audit table to understand what that rule is likely to do operationally.

This viewpoint also sharpens a broader conceptual point. A conformal output is not fully characterized, for deployment purposes, by its marginal coverage or by its set-valued form alone. What matters operationally is how calibrated regions align with labels in the deployment distribution. That is why coverage-matched settings can still differ in commitment, deferral, decisive error exposure, and downstream cost compatibility. In this sense, the region–class table is not auxiliary bookkeeping; it is the minimal object in our setup that connects a fixed conformal interface to operational and decision-theoretic consequences. This also gives a concrete response to the HCI concern that conformal sets can be a "murky" interface: the set alone is often not enough, but the set paired with its audited region–class table makes the deployment-facing consequences of acting on that interface explicit.

The scope remains intentionally narrow. We treat the score model as fixed, focus on binary classification, and rely on exchangeability. The exact finite-sample claims apply to SSBC's coverage semantics and to independent-audit inference for fixed operational rates; the LOO interface is used only as a planning proxy, and attainable-set sweeps and Pareto-front views are exploratory rather than simultaneous certification statements. Extending the same viewpoint to multi-class outputs, structured predictions, richer abstention policies, adaptive policies, or shift-aware settings remains future work. The most serious practical limitation is distribution or data drift: once exchangeability fails, the finite-sample guarantees need not hold, so deployment would require monitoring, recalibration, or shift-aware extensions (Fannjiang et al., 2022). Our aim is to make the fixed-rule deployment picture explicit before those complications are layered in.

## Acknowledgements

Removed for deidentification purposes.

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

# A    Appendix Outline & Notation

This appendix serves as a notation guide and roadmap for the technical appendices that follow. The appendices are, in order:

- **A. Appendix Outline & Notation** (this appendix): roadmap and shared notation for calibration, SSBC, and operational analysis.

- **B. Finite-sample distribution of calibration-conditional coverage:** pivot and distribution of coverage given the calibration draw; basis for coverage-conditional guarantees.

- **C. Small-Sample Beta Correction (SSBC):** mapping $(\alpha^\star, \delta)$ to a calibration grid index; finite-sample coverage semantics for the deployed rule.

- **D. Single-sample structural coupling, LOO decoupling, and planning envelopes:** why an independent audit split is needed for certification; leave-one-out surrogate and envelope construction when no audit split is available.

- **E. Binary conformal partitions:** four-region structure, regime boundary $(\tau_0 + \tau_1 = 1)$, and recovery of the argmax and Bayes optimal classification rule under asymmetric costs.

- **F. Conformal interface-relative decision optimality and inverse pricing:** cost-ratio conditions for decision-optimal action; Chow-style accept/reject and pricing envelopes.

- **G. Explicit region indicators and projection masks:** formulas for region–label counts and projections used in auditing and KPI computation.

- **H. Tox21 supplementary details:** dataset, splits, and operational summaries for the Tox21 empirical example.

- **I. Solubility supplementary details:** dataset, scenario restriction, Pareto-front parameter roles, and $\alpha$ vs. $\delta$ interpretation for the solubility case study.

The notation below is shared across split conformal calibration, the Small-Sample Beta Correction (SSBC), and finite-window prediction of operational rates. The analysis mixes three layers: (i) discrete split-conformal thresholding via order statistics on calibration scores; (ii) SSBC grid selection and calibration-conditional coverage semantics; and (iii) predictive envelopes for deployment-facing operational rates (Calibrate-and-Audit and a single-sample leave-one-out (LOO) surrogate). To avoid collisions, we reserve $(k, u)$ exclusively for split conformal thresholding. Table 2 summarizes the key notation.

Table 2: Key notation used across calibration, SSBC, and operational analysis.

| Symbol | Meaning | Scope / Remarks |
|---|---|---|
| *Data splits and window sizes* | | |
| $\mathcal{D}_{\mathrm{cal}}$, $n_{\mathrm{cal}}$ | Calibration dataset and size | Exchangeable sample for conformal thresholds. |
| $\mathcal{D}_{\mathrm{audit}}$, $n_{\mathrm{audit}}$ | Audit dataset and size | Exchangeable sample for region–class label and operational rate estimation. |
| $m$ | Future window size | Number of future cases for realized rates and predictive envelopes. |
| *Region–policy–audit* | | |
| $R_\tau(x)$, $R_{\tau(\theta)}$ | Region map | Partition of score space; $R_\tau : \mathcal{X} \to \{0,1\}^K$; fixed by thresholds $\tau(\theta)$. |
| $\pi$ | Deployment policy | Maps region $R_\tau(x)$ to reported output (e.g., prediction set or $\varnothing$). |
| $\theta$, $\Theta$ | Calibration setting and space | Indexes deployed thresholds $\tau(\theta)$ and region map $R_{\tau(\theta)}$. |
| *Split conformal and SSBC* | | |
| $s(x,y)$, $S_i$, $S_{(k)}$ | Score, calibration scores, order statistic | $\tau = S_{(k)}$; $k, u$ with $u = n_{\mathrm{cal}} + 1 - k$ index the conformal grid. |
| $\alpha_{\mathrm{grid}}$, $\alpha_{\mathrm{adj}}$ | Grid and SSBC-selected miscoverage | $\alpha_{\mathrm{grid}} = u/(n_{\mathrm{cal}} + 1)$; SSBC returns $\alpha_{\mathrm{adj}} = u^\star/(n_{\mathrm{cal}} + 1)$. |
| $\alpha^\star$, $\delta$ | Target miscoverage, confidence | User request; $\delta$ controls tail probability over calibration draws. |
| $p_{\mathrm{cov}}$, $\widehat{C}_m$ | Coverage probability, empirical coverage | $p_{\mathrm{cov}} \sim \mathrm{Beta}(k, u)$; $\widehat{C}_m = \frac{1}{m} \sum_{j=1}^m \mathbf{1}\{Y_j' \in C(X_j')\}$. |
| *Operational indicators and LOO* | | |
| $C(X)$, $g_\ell$, $I_\ell$ | Prediction set, event functional, indicator | $I_\ell = g_\ell(R_{\tau(\theta)}(X), Y; \pi)$; KPIs are sums over selected region–class cells. |
| $Z_{i,j}$, $k_{\mathrm{pool},j}$, $\widehat{r}_j^{\mathrm{LOO}}$ | LOO indicator, pooled count, LOO rate | $Z_{i,j} = g_j(C_{-i}(X_i), Y_i)$; $k_{\mathrm{pool},j} = \sum_i Z_{i,j}$; $\widehat{r}_j^{\mathrm{LOO}} = k_{\mathrm{pool},j}/n$. |
| `infl`, $n_{\mathrm{eff}}$ | Inflation (pessimization) | $n_{\mathrm{eff}} = n/\mathtt{infl}$ widens LOO envelopes (Appendix D); larger `infl` yields wider intervals. |

# B  Finite-sample distribution of calibration-conditional coverage

This appendix derives a finite-sample characterization of the *calibration-conditional coverage* of a fixed split conformal predictor under exchangeability. Coverage is a pure *rank* event: the future true-label score is compared to a calibration order statistic. This yields a distribution-free pivot and a Beta law for realized (calibration-conditional) coverage across calibration draws. This pivot is the input to SSBC (Appendix C). The derivation presented here is a one-shot rank/order-statistic pivot, complementary to the Beta–Binomial/de Finetti route in Marques Filho's analysis of the exchangeable sequence of future coverage indicators (Marques, 2025).

## B.1  Setup

Let $\mathcal{D}_{\mathrm{cal}} = \{(X_i, Y_i)\}_{i=1}^n$ be exchangeable with a future test pair $(X_{n+1}, Y_{n+1})$. Let $s(x, y)$ be a nonconformity score and define

$$S_i := s(X_i, Y_i), \quad i = 1, \ldots, n, \qquad \text{and} \qquad S_{n+1} := s(X_{n+1}, Y_{n+1}).$$

Fix an index $k \in \{1, \ldots, n\}$ and set

$$\tau := S_{(k)}, \qquad u := n + 1 - k, \qquad \alpha_{\mathrm{grid}} = \frac{u}{n+1}.$$

For the predictor calibrated on $\mathcal{D}_{\mathrm{cal}}$, the *calibration-conditional coverage probability* is

$$p_{\mathrm{cov}}(\mathcal{D}_{\mathrm{cal}}) := \mathbb{P}(S_{n+1} \le \tau \mid \mathcal{D}_{\mathrm{cal}}).$$

This random variable varies across calibration draws but is fixed once calibration is completed.

## B.2  Rank pivot

Assume for clarity that the score distribution is continuous so ties occur with probability zero (ties are addressed in Remark 1). Under exchangeability of $\{S_1, \ldots, S_n, S_{n+1}\}$, the rank

$$R := \mathrm{rank}\left(S_{n+1} \text{ among } S_1, \ldots, S_n, S_{n+1}\right)$$

is uniform on $\{1, \ldots, n+1\}$ (David & Nagaraja, 2003). Since $\tau = S_{(k)}$,

$$\{S_{n+1} \le \tau\} \quad \Longleftrightarrow \quad \{R \le k\}.$$

Thus coverage is the conditional probability of a rank event.

## B.3  Beta law

A standard order-statistic identity implies that the conditional probability of the rank event equals the $k$th order statistic of $n + 1$ i.i.d. uniforms:

$$p_{\mathrm{cov}}(\mathcal{D}_{\mathrm{cal}}) \overset{d}{=} U_{(k)}, \qquad U_{(k)} \sim \mathrm{Beta}(k, u), \quad u = n + 1 - k,$$

where $U_{(k)}$ is the $k$th order statistic of $U_1, \ldots, U_{n+1} \overset{\text{i.i.d.}}{\sim} \mathrm{Unif}(0, 1)$ (David & Nagaraja, 2003). Equivalently, for any $t \in [0, 1]$,

$$\mathbb{P}(p_{\mathrm{cov}}(\mathcal{D}_{\mathrm{cal}}) \le t) = I_t(k, u),$$

where $I_t(\cdot, \cdot)$ is the regularized incomplete Beta function. The law is distribution-free and depends only on $(n, k)$ (equivalently $(n, u)$).

*Remark* 1 (Discrete scores and ties). If scores have atoms, ranks are not almost surely unique. Exact pivots can be recovered by randomized tie-breaking; deterministic left/right-continuous conventions yield conservative bounds. The non-interpolated order-statistic thresholding convention used in the main text preserves finite-sample validity.

Coverage depends only on the rank of the future true-label score relative to the calibration scores, hence admits a finite-sample distribution-free law. Most other operational KPIs depend on the joint geometry of multiple label scores and threshold interactions and therefore do not admit an analogous pivot (Appendix D).

# C   Small-Sample Beta Correction (SSBC)

SSBC is a deterministic index-selection rule for split conformal calibration that makes a user request $(\alpha^\star, \delta)$ operationally precise for the *single deployed predictor* obtained after one calibration draw. SSBC selects a conformal grid index (equivalently an order statistic) so that, with probability at least $1 - \delta$ over calibration randomness, the realized coverage of the deployed rule is at least $1 - \alpha^\star$. In the finite-window variant, the same guarantee is imposed on empirical coverage over a future window of size $m$. The construction relies only on the finite-sample distribution of calibration-conditional coverage from Appendix B.

## C.1   Setup and conformal grid

Let $n_{\mathrm{cal}}$ be the calibration size and let $S_i := s(X_i, Y_i)$ denote true-label nonconformity scores for $i = 1, \ldots, n_{\mathrm{cal}}$. Split conformal selects a threshold as an order statistic:

$$\tau = S_{(k)}, \qquad k \in \{1, \ldots, n_{\mathrm{cal}}\}.$$

It is convenient to re-index the same grid by the miscoverage index

$$u := n_{\mathrm{cal}} + 1 - k \in \{1, \ldots, n_{\mathrm{cal}}\}, \qquad \alpha_{\mathrm{grid}} = \frac{u}{n_{\mathrm{cal}} + 1}, \qquad k = n_{\mathrm{cal}} + 1 - u.$$

Here $\alpha^\star \in (0, 1)$ is the requested miscoverage and $\delta \in (0, 1)$ is a confidence/risk level controlling the probability (over calibration draws) that the requested semantics fail.

## C.2   Distribution of realized coverage

For the fixed predictor calibrated on $\mathcal{D}_{\mathrm{cal}}$, define the calibration-conditional coverage probability

$$p_{\mathrm{cov}}(\mathcal{D}_{\mathrm{cal}}) := \mathbb{P}(S_{n+1} \leq \tau \mid \mathcal{D}_{\mathrm{cal}}),$$

where $(X_{n+1}, Y_{n+1})$ is exchangeable with the calibration sample. Under exchangeability, Appendix B shows that

$$p_{\mathrm{cov}}(\mathcal{D}_{\mathrm{cal}}) \sim \mathrm{Beta}(k, u), \qquad k = n_{\mathrm{cal}} + 1 - u,$$

exactly and distribution-free. This describes how the realized coverage of the deployed predictor varies across hypothetical recalibrations, while treating the deployed rule as fixed after the one calibration step.

## C.3   SSBC objective: calibration-conditional PAC semantics

SSBC enforces the calibration-conditional PAC-style constraint

$$\mathbb{P}_{\mathcal{D}_{\mathrm{cal}}}(p_{\mathrm{cov}}(\mathcal{D}_{\mathrm{cal}}) \geq 1 - \alpha^\star) \geq 1 - \delta.$$

Using the Beta law, this is equivalent to

$$\mathbb{P}(Z \geq 1 - \alpha^\star) \geq 1 - \delta, \qquad Z \sim \mathrm{Beta}(k, u).$$

**Selection rule: least conservative admissible grid point.** Among discrete grid indices $u \in \{1, \ldots, n_{\mathrm{cal}}\}$, SSBC selects the *largest admissible $u$* satisfying the tail constraint above:

$$u^\star := \max\left\{ u \in \{1, \ldots, n_{\mathrm{cal}}\} : \ \mathbb{P}(Z \geq 1 - \alpha^\star) \geq 1 - \delta, \ Z \sim \mathrm{Beta}(n_{\mathrm{cal}} + 1 - u, \ u) \right\}.$$

Equivalently, SSBC deploys the least conservative grid miscoverage level $\alpha_{\mathrm{adj}} = u^\star/(n_{\mathrm{cal}} + 1)$ that certifies the requested semantics. The returned order-statistic index is $k_{\mathrm{adj}} = n_{\mathrm{cal}} + 1 - u^\star$.

### C.4 Finite-window deployment semantics

In many deployments, coverage is evaluated over a finite window of size $m$. Define empirical coverage over that window by

$$\widehat{C}_m := \frac{1}{m}\sum_{j=1}^{m} I_{\mathrm{cov},j}, \qquad I_{\mathrm{cov},j} := \mathbf{1}\{Y_j' \in C(X_j')\}, \qquad S_m := m\widehat{C}_m.$$

Conditional on the calibration-conditional coverage probability $p_{\mathrm{cov}}(\mathcal{D}_{\mathrm{cal}}) = p$,

$$S_m \mid p \sim \mathrm{Binomial}(m, p).$$

Marginalizing $p \sim \mathrm{Beta}(k, u)$ yields the mixture distribution

$$S_m \sim \text{Beta-Binomial}(m; k, u), \qquad k = n_{\mathrm{cal}} + 1 - u.$$

The finite-window SSBC criterion selects $u$ such that

$$\mathbb{P}\left(\widehat{C}_m \geq 1 - \alpha^\star\right) \ \geq \ 1 - \delta,$$

where the probability is over both calibration randomness and the future window.

#### C.4.1 Strict lower-tail convention

Because $\widehat{C}_m$ is discrete, we adopt a strict violation convention $\widehat{C}_m < 1 - \alpha^\star$. Define the corresponding count threshold

$$x^\star := \lfloor (1 - \alpha^\star)m \rfloor + 1, \qquad \text{so that} \qquad \{\widehat{C}_m \geq 1 - \alpha^\star\} \iff \{S_m \geq x^\star\}.$$

All Beta–Binomial tail probabilities in SSBC are evaluated for the event $\{S_m \geq x^\star\}$. This avoids boundary ambiguity when $(1 - \alpha^\star)m$ is an integer and preserves monotonicity in $u$.

### C.5 Infinite-window limit

This limiting regime is identified by Marques (2025). In our notation, if $u_m$ denotes the SSBC-selected grid index obtained by inverting the Beta–Binomial tail for window size $m$, and $u_\infty$ denotes the index obtained from the Beta limit law, then $u_m \to u_\infty$ as $m \to \infty$. Intuitively, conditional on $p_{\mathrm{cov}} = p$, the empirical coverage $\widehat{C}_m$ converges almost surely to $p$, so the finite-window Beta–Binomial criterion approaches the infinite-window Beta criterion.

### C.6 Feasibility and saturation

Not every pair $(\alpha^\star, \delta)$ is feasible at fixed $n_{\mathrm{cal}}$ because calibration choices lie on the conformal grid. Under the most conservative grid point $u = 1$ (equivalently $k = n_{\mathrm{cal}}$),

$$p_{\mathrm{cov}} \sim \mathrm{Beta}(n_{\mathrm{cal}}, 1), \qquad \mathbb{P}(p_{\mathrm{cov}} \geq 1 - \alpha) = 1 - (1 - \alpha)^{n_{\mathrm{cal}}}.$$

Thus any infinite-window PAC requirement at confidence $1 - \delta$ must satisfy

$$\alpha \ \geq \ 1 - \delta^{1/n_{\mathrm{cal}}}.$$

If this condition is violated (and likewise in the finite-window analogue), no grid point can satisfy the tail constraint and SSBC returns INFEASIBLE.

### C.7 Coverage semantics under finite calibration

To visualize how nominal requests map to deployed semantics under finite calibration, Figure 6 plots the effective calibration level $\alpha_{\mathrm{adj}}$ selected by SSBC as a function of the user inputs $(\alpha, \delta)$ at fixed $n_{\mathrm{cal}}$. Distinct nominal requests can induce the same deployed grid index and therefore the same realized coverage semantics.

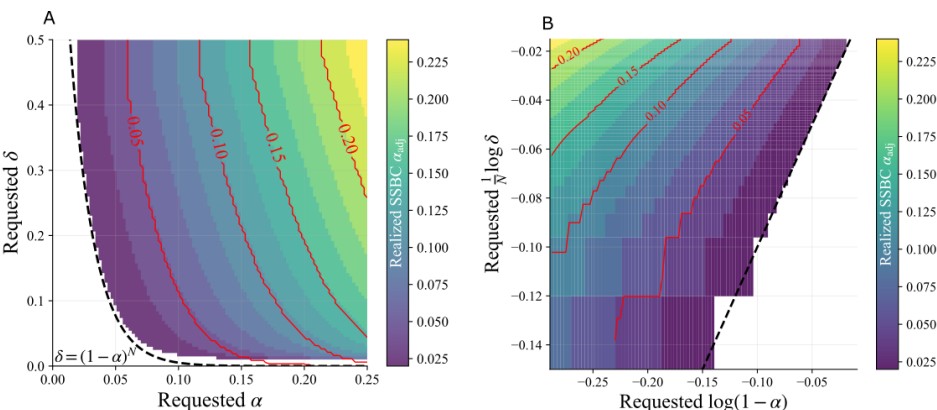

Figure 6: **Semantic interpretation of nominal coverage requests under finite calibration.** Each panel visualizes the effective calibration level $\alpha_{\text{adj}}$ selected by SSBC as a function of the user-specified miscoverage level $\alpha$ and confidence level $\delta$, for fixed $n_{\text{cal}}$. Color encodes the deployed level $\alpha_{\text{adj}}$, while contours indicate iso-semantic sets: distinct nominal requests that induce the same deployed calibration grid point. The feasibility boundary reflects the finite-sample constraint $\alpha \gtrsim 1 - \delta^{1/n_{\text{cal}}}$.

### C.8 SSBC algorithm (deterministic specification)

This subsection provides a reproducible implementation-level specification. The algorithm returns the largest admissible $u$ (least conservative grid point) satisfying the relevant tail constraint.

---

**Algorithm 1** Small-Sample Beta Correction (SSBC)

---

**Require:** Target miscoverage $\alpha^\star \in (0, 1)$; calibration size $n_{\text{cal}} \in \mathbb{N}$; confidence $\delta \in (0, 1)$; deployment regime $\in \{\infty, m\}$ (window size $m$ if finite)

**Ensure:** Adjusted grid level $\alpha_{\text{adj}}$ and index $k_{\text{adj}}$, or INFEASIBLE

1: $t \leftarrow 1 - \alpha^\star$
2: $u^\star \leftarrow -1$
3: **if** regime $= m$ **then**
4:      $x^\star \leftarrow \lfloor t\,m \rfloor + 1$
5: **end if**
6: **for** $u = 1, \ldots, n_{\text{cal}}$ **do**
7:      $a \leftarrow n_{\text{cal}} + 1 - u, \quad b \leftarrow u$
8:      **if** regime $= \infty$ **then**
9:          $p_{\text{tail}} \leftarrow \Pr[Z \geq t], \ Z \sim \text{Beta}(a, b)$
10:      **else**
11:          $p_{\text{tail}} \leftarrow \Pr[X \geq x^\star], \ X \sim \text{Beta-Binomial}(m; a, b)$
12:      **end if**
13:      **if** $p_{\text{tail}} \geq 1 - \delta$ **then**
14:          $u^\star \leftarrow u$
15:      **end if**
16: **end for**
17: **if** $u^\star < 0$ **then**
18:      **return** INFEASIBLE
19: **end if**
20: $\alpha_{\text{adj}} \leftarrow \frac{u^\star}{n_{\text{cal}}+1}$
21: $k_{\text{adj}} \leftarrow n_{\text{cal}} + 1 - u^\star$
22: **return** $\alpha_{\text{adj}}, \ k_{\text{adj}}$

---

**Relation to DKWM-style calibration.** DKWM-style calibration modifies the nominal grid choice to enforce conservative, worst-case guarantees uniformly over calibration draws and distributions. SSBC addresses a different question: it assigns a calibration-conditional PAC meaning to a user request $(\alpha^\star, \delta)$ for the *single deployed* predictor produced after one calibration. DKWM targets uniform validity across hypothetical recalibrations; SSBC targets admissibility of the realized rule via Beta (or Beta–Binomial) tails.

### C.9 Extended simulation table for calibration-conditional violations

Table 3 reports the full calibration-size grid for the simulation summarized in Section 5.1.1.

Table 3: **Calibration-conditional coverage violation rates with theory.** Target miscoverage $\alpha^\star = 0.10$, confidence $\delta = 0.10$. $\alpha_{\mathrm{grid}}$ is the grid point selected on the conformal grid, $\alpha_{\mathrm{cont}}$ is the requested value under the DKWM correction, and $m_{\mathrm{cal}}$ is the calibration-window size. Results are based on $10^6$ calibration draws and a finite deployment window of size $m_{\mathrm{infer}} = 100$. The **Obs** column reports $\mathbb{P}(\widehat{C}_m < 1 - \alpha^\star)$. **Beta** reports $\mathbb{P}(p_{\mathrm{cov}} < 1 - \alpha^\star)$ under $p_{\mathrm{cov}} \sim \mathrm{Beta}(k, u)$. **BetaBinom** reports $\mathbb{P}(\widehat{C}_m < 1 - \alpha^\star)$ under the induced Beta–Binomial law.

| $n_{\mathrm{cal}}$ | Method | $u$ | $\alpha_{\mathrm{grid}}$ | Obs | Beta | BetaBinom | $\alpha_{\mathrm{cont}}$ |
|---|---|---|---|---|---|---|---|
| 50 | None | 5 | 0.0980 | 0.3963 | 0.4312 | 0.3964 | – |
| 50 | SSBC | 2 | 0.0392 | 0.0476 | 0.0338 | 0.0472 | – |
| 50 | DKWM | 1 | 0.0196 | 0.0096 | 0.0052 | 0.0095 | $-0.0731$ |
| 75 | None | 7 | 0.0921 | 0.3454 | 0.3673 | 0.3464 | – |
| 75 | SSBC | 4 | 0.0526 | 0.0769 | 0.0504 | 0.0768 | – |
| 75 | DKWM | 1 | 0.0132 | 0.0016 | 0.0004 | 0.0017 | $-0.0413$ |
| 100 | None | 10 | 0.0990 | 0.4075 | 0.4513 | 0.4071 | – |
| 100 | SSBC | 6 | 0.0594 | 0.0960 | 0.0576 | 0.0956 | – |
| 100 | DKWM | 1 | 0.0099 | 0.0004 | 0.0000 | 0.0004 | $-0.0224$ |
| 150 | None | 15 | 0.0993 | 0.4119 | 0.4602 | 0.4107 | – |
| 150 | SSBC | 9 | 0.0596 | 0.0804 | 0.0307 | 0.0801 | – |
| 150 | DKWM | 1 | 0.0066 | 0.0000 | 0.0000 | 0.0000 | $+0.0001$ |
| 200 | None | 20 | 0.0995 | 0.4130 | 0.4655 | 0.4124 | – |
| 200 | SSBC | 13 | 0.0647 | 0.0974 | 0.0320 | 0.0980 | – |
| 200 | DKWM | 2 | 0.0100 | 0.0000 | 0.0000 | 0.0000 | $+0.0135$ |
| 250 | None | 25 | 0.0996 | 0.4126 | 0.4692 | 0.4134 | – |
| 250 | SSBC | 16 | 0.0637 | 0.0863 | 0.0175 | 0.0858 | – |
| 250 | DKWM | 5 | 0.0199 | 0.0003 | 0.0000 | 0.0003 | $+0.0226$ |
| 300 | None | 30 | 0.0997 | 0.4139 | 0.4719 | 0.4141 | – |
| 300 | SSBC | 20 | 0.0664 | 0.0969 | 0.0171 | 0.0971 | – |
| 300 | DKWM | 8 | 0.0266 | 0.0009 | 0.0000 | 0.0009 | $+0.0293$ |
| 500 | None | 50 | 0.0998 | 0.4153 | 0.4782 | 0.4153 | – |
| 500 | SSBC | 34 | 0.0679 | 0.0949 | 0.0049 | 0.0955 | – |
| 500 | DKWM | 22 | 0.0439 | 0.0097 | 0.0000 | 0.0096 | $+0.0453$ |

# D  Single-sample structural coupling, leave-one-out decoupling, and planning-envelope inflation

The two-stage predictive reference used throughout this paper separates *calibration* (choose a conformal threshold on $\mathcal{D}_{\mathrm{cal}}$) from *operational evaluation* (estimate rates on an independent window). This separation is what makes window indicators behave as i.i.d. Bernoulli draws under a fixed deployed rule.

In practice, an independent audit set is not always available. When the same sample is reused both to select the threshold and to estimate operational rates, threshold selection and evaluation become coupled. This appendix records (i) a minimal structural reason for reuse-induced dependence, (ii) a data-efficient remedy, leave-one-out (LOO) recalibration, that provides effective *practical decoupling* in our regimes, and (iii) an *inflation* parameter `infl` that pessimizes predictive envelopes when residual dependence or regime instability remains. Throughout, these LOO constructions are intended for exploratory planning when a dedicated audit split is unavailable, we cannot provide strong theoretical guarantees on the rates of the LOO proxy indicators at this point.

## D.1  Why single-sample reuse introduces dependence

Let $\mathcal{D}_{\mathrm{cal}} = \{(X_i, Y_i)\}_{i=1}^n$ be an exchangeable calibration sample, let $S_i := s(X_i, Y_i)$ be true-label nonconformity scores, and let the split conformal threshold be the $k$th order statistic

$$\hat{\tau} := S_{(k)}, \qquad k \in \{1, \dots, n\},$$

assuming continuity so ties occur with probability zero. For any threshold $t$, define the crossing indicator $I_i(t) := \mathbf{1}\{S_i \le t\}$.

**Insight from counting.**  If $\hat{\tau} = t$ is the $k$th order statistic, then exactly $k - 1$ scores lie strictly below $t$ and one score equals $t$ (under continuity, almost surely), so

$$\sum_{i=1}^n I_i(t) = k \quad \text{almost surely.}$$

This already shows that the indicators $\{I_i(t)\}_{i=1}^n$ cannot be conditionally independent given $\hat{\tau} = t$: once some are known to be one, fewer ones remain available for the others. By exchangeability, for any $i \ne j$,

$$\mathrm{Cov}(I_i(t), I_j(t) \mid \hat{\tau} = t) := \frac{k(k-n)}{n^2(n-1)} < 0 \quad (k < n).$$

**Implication for operational envelopes.**  Operational indicators are functions of the deployed prediction set $C_{\hat{\tau}}(X) = \{y : s(X, y) \le \hat{\tau}\}$ and therefore inherit reuse-induced dependence. In particular, the nonzero conditional covariance shows that these indicators are not Bernoulli draws under a fixed rule once threshold selection and evaluation are performed on the same sample. To recover the Bernoulli sampling picture needed for the two-stage guarantees in the main text, one therefore needs an independent audit set after calibration. If one naïvely treats reuse-based indicators as i.i.d. Bernoulli under a fixed rule, predictive envelopes can become under-dispersed.

## D.2  Approximate decoupling via leave-one-out (LOO)

When an independent audit sample is unavailable, we use LOO recalibration (Vovk, 2015; Barber et al., 2021) to reduce self-influence. Let $\hat{\tau}_{-i}$ be the split conformal threshold computed on $\mathcal{D}_{\mathrm{cal}} \setminus \{(X_i, Y_i)\}$, and let $C_{-i}(\cdot)$ be the corresponding prediction set map. For an operational event functional $g_j(C(X), Y) \in \{0, 1\}$ (e.g., singleton, doublet, wrong-singleton), define LOO indicators

$$Z_{i,j} := g_j(C_{-i}(X_i), Y_i), \qquad i = 1, \dots, n,$$

and pooled LOO summaries

$$k_{\mathrm{pool},j} := \sum_{i=1}^n Z_{i,j}, \qquad \hat{r}_j^{\mathrm{LOO}} := \frac{k_{\mathrm{pool},j}}{n}.$$

**Proxy two-stage interpretation.** Each $Z_{i,j}$ is evaluated under a rule that does not use point $i$, restoring a localized separation between rule construction and evaluation. Although the rule varies across folds, each fold differs only slightly from the full calibrated rule, so $\{Z_{i,j}\}$ can be read as indicators from nearby operating regimes. Pooling provides a direct empirical proxy for operational behavior under finite calibration.

**Empirical decoupling.** In the regimes studied (Section 5.1.2), LOO envelopes track the two-stage Calibrate–and–Audit reference closely in center and often slightly pessimistically in width. We therefore use pooled LOO indicators for planning when a separate audit set is unavailable.

### D.3 Envelope inflation as controlled pessimization

LOO is local and does not remove all dependence, especially near regime boundaries where small threshold shifts can change region support. We therefore introduce an explicit inflation parameter `infl` $\geq 1$ that widens predictive envelopes by shrinking an effective sample size used in the predictive model.

**LOO-intrinsic pessimization compounded by SSBC.** LOO recalibration introduces an intrinsic pessimization because each fold threshold $\hat{\tau}_{-i}$ is computed on only $n-1$ calibration points. On top of this, we apply the small-sample Beta correction (SSBC) to achieve the desired nominal level $\alpha$ under finite calibration uncertainty. The SSBC step further increases conservatism (i.e., widens envelopes) relative to a plug-in rate estimate, so the combined LOO+SSBC construction is typically wider than either component alone. Importantly, SSBC is applied using the *actual* fold sample size (e.g., $n-1$) to correct the nominal tail level, whereas `infl` below is a separate width knob acting only through the downstream dispersion model (we do not apply SSBC to $n_{\text{eff}}$).

**Rank stability of LOO thresholds ($\pm 1$ order statistic).** In split conformal, the threshold is an order statistic of the calibration scores. Let $S_1, \ldots, S_n$ be the full-sample calibration scores with order statistics $S_{(1)} \leq \cdots \leq S_{(n)}$, and let $k := \lceil (n+1)(1-\alpha) \rceil$ so that $\hat{\tau} = S_{(k)}$. Under LOO, the fold uses $n-1$ scores and index $k' := \lceil n(1-\alpha) \rceil \in \{k-1, k\}$. Removing a single score can shift the rank position of the selected order statistic by at most one, hence (ignoring ties) the LOO threshold satisfies

$$\hat{\tau}_{-i} \in \{S_{(k-1)}, S_{(k)}, S_{(k+1)}\},$$

and more generally lies between adjacent full-sample order statistics. This formalizes the "local" nature of LOO recalibration, while still allowing support changes near regime boundaries when $C(\cdot)$ is sensitive to small threshold movements.

**Operational effect.** In constructions below we replace a nominal proxy sample size $n$ by $n_{\text{eff}} = n/\texttt{infl}$ (and analogously for any proxy count used to parameterize predictive dispersion). Larger `infl` yields wider envelopes without changing the pooled mean, providing a monotone knob for conservatism.

**Diagnostic guidance (variance-based inflation).** As a diagnostic of calibration-induced variability, compute fold rates $\hat{r}^{(-i)}$ under each LOO-calibrated rule and their empirical variance

$$\bar{r}_{\text{LOO}} = \frac{1}{n} \sum_{i=1}^{n} \hat{r}^{(-i)}, \qquad \widehat{\text{Var}}_{\text{LOO}} = \frac{1}{n-1} \sum_{i=1}^{n} \left( \hat{r}^{(-i)} - \bar{r}_{\text{LOO}} \right)^2.$$

To isolate variability attributable to *recalibration* (rule changes) from simple delete-one evaluation noise, we also form a *not-LOO* reference in which the operating rule is held fixed. Let

$$W_i := g_j(C(X_i), Y_i), \qquad \hat{r}^{\text{full}} := \frac{1}{n} \sum_{i=1}^{n} W_i,$$

and define delete-one evaluation rates under the fixed full rule $C(\cdot)$ by

$$\hat{r}^{\text{full}(-i)} := \frac{1}{n-1} \sum_{\ell \neq i} W_\ell = \frac{n\,\hat{r}^{\text{full}} - W_i}{n-1}, \qquad i = 1, \ldots, n.$$

Let $\widehat{\mathrm{Var}}_{\mathrm{full}}$ denote the empirical variance of $\{\widehat{r}^{\mathrm{full}(-i)}\}_{i=1}^n$. We then define a variance ratio

$$q := \frac{\widehat{\mathrm{Var}}_{\mathrm{LOO}}}{\widehat{\mathrm{Var}}_{\mathrm{full}}}$$

and use it to suggest an inflation level

$$\mathtt{infl} := \max\{1, q\},$$

Intuitively, $\widehat{\mathrm{Var}}_{\mathrm{full}}$ measures baseline delete-one variability when the rule is fixed, while $\widehat{\mathrm{Var}}_{\mathrm{LOO}}$ captures additional spread induced by recalibration. Although $\widehat{\tau}_{-i}$ can move by at most one rank, the resulting change in region support (and therefore in $g_j$) can be discontinuous near regime boundaries; larger $q$ flags such sensitivity and points toward higher $\mathtt{infl}$.

In our experiments the variance ratio is typically only slightly above 1, indicating that LOO recalibration contributes a modest amount of additional variability beyond delete-one evaluation under a fixed rule. This is consistent with the rank-1 stability of LOO thresholds (up to ties).

### D.4 Predictive envelope constructions from LOO proxies

We describe two complementary envelope constructions built from LOO proxy indicators. The first mirrors the two-stage reference and is the primary approximation; the second is a conservative guardrail.

**Beta–Binomial planning envelopes.** Let $Z_i$ denote a pooled proxy indicator for a fixed KPI (suppress $j$), and let $k_{\mathrm{pool}} = \sum_{i=1}^n Z_i$ with pooled proxy rate $\widehat{p} = k_{\mathrm{pool}}/n$. Apply inflation via

$$n_{\mathrm{eff}} = \frac{n}{\mathtt{infl}}, \qquad k_{\mathrm{eff}} = \widehat{p}\, n_{\mathrm{eff}} = \frac{k_{\mathrm{pool}}}{\mathtt{infl}}.$$

With a small prior offset $\mathrm{offset} \in \{1, 1/2\}$, define

$$\alpha := k_{\mathrm{eff}} + \mathrm{offset}, \qquad \beta := (n_{\mathrm{eff}} - k_{\mathrm{eff}}) + \mathrm{offset}.$$

For a future deployment window of size $m$, the predictive count $S_m$ is modeled as

$$S_m \sim \mathrm{BetaBinomial}(m, \alpha, \beta),$$

and equal-tailed prediction intervals follow from the Beta–Binomial CDF. Increasing $\mathtt{infl}$ shrinks $n_{\mathrm{eff}}$ and widens intervals monotonically while preserving the pooled mean.

**Hoeffding-type dominance bound.** As a conservative alternative, Hoeffding's inequality gives a distribution-free bound for a window rate $\widehat{r}_m$ around a proxy mean $\widehat{r}_{\mathrm{LOO}}$:

$$\Pr(|\widehat{r}_m - \widehat{r}_{\mathrm{LOO}}| \geq \epsilon) \leq 2\exp(-2m\epsilon^2),$$

yielding simple symmetric envelopes that can be used as a worst-case planning check.

**Summary.** Single-sample reuse induces structural dependence through the thresholding mechanism described above, primarily distorting predictive *dispersion*. LOO recalibration reduces self-influence and closely tracks audit-style behavior in our regimes. Predictive envelopes are then constructed from pooled LOO proxies using a Beta–Binomial model, with $\mathtt{infl}$ providing a monotone knob for controlled pessimization and the Hoeffding bound serving as a conservative guardrail. At this point, we cannot provide strong theoretical guarantees on the rates of the LOO proxy indicators, and therefore recommend using a proper audit set for certification.

## E  Binary conformal partitions: regimes, coupling, and rate primitives

This appendix records the geometric facts used in Section 2, Section 3, and Section 4. Here we assume thresholds are fixed and all probabilities are taken with respect to the deployment distribution conditional on $\mathcal{D}_{\mathrm{cal}}$.

### E.1 Class-conditional split conformal as a four-region partition.

Let $\mathcal{Y} = \{0, 1\}$ and let $s(x, y)$ be a nonconformity score. Class-conditional split conformal produces thresholds $\tau_0, \tau_1$, and the set-valued output is

$$\mathcal{C}(x) = \{0 : s(x, 0) \leq \tau_0\} \cup \{1 : s(x, 1) \leq \tau_1\}, \tag{2}$$

equivalently represented by the region label

$$R_\tau(x) := \big(\mathbf{1}\{s(x, 0) \leq \tau_0\},\ \mathbf{1}\{s(x, 1) \leq \tau_1\}\big) \in \{0, 1\}^2, \qquad \tau = (\tau_0, \tau_1).$$

Writing $(s_0, s_1) = (s(x, 0), s(x, 1))$, the thresholds partition score space into

$$R_\tau(x) = \begin{cases} 11, & s_0 \leq \tau_0,\ s_1 \leq \tau_1 \quad \text{(doublet)}, \\ 10, & s_0 \leq \tau_0,\ s_1 > \tau_1 \quad \text{(singleton } \{0\}), \\ 01, & s_0 > \tau_0,\ s_1 \leq \tau_1 \quad \text{(singleton } \{1\}), \\ 00, & s_0 > \tau_0,\ s_1 > \tau_1 \quad \text{(abstention)}. \end{cases}$$

For any fixed $\tau$,

$$\sum_{r \in \{00, 01, 10, 11\}} \mu_r(\tau) = 1, \qquad \mu_r(\tau) := \Pr(R_\tau(X) = r \mid \mathcal{D}_{\text{cal}}).$$

### E.2 Probability-normalized scores and a sharp regime boundary

Many probabilistic classifiers induce probability-normalized scores, e.g. $s(x, y) = 1 - P(y \mid x)$. For $\mathcal{Y} = \{0, 1\}$ this implies

$$s(x, 0) + s(x, 1) = 1,$$

so feasible score pairs lie on the diagonal manifold

$$\mathcal{M} = \{(u, 1 - u) : u \in [0, 1]\}.$$

Intersecting $\mathcal{M}$ with the threshold rectangles yields a sharp boundary that determines which region types can occur with nonzero mass.

**Proposition 2** (Regime boundary under probability normalization). *Assume $(s_0, s_1) \in \mathcal{M}$ almost surely. Then:*

1. *$R_{11}$ has nonempty intersection with $\mathcal{M}$ iff $\tau_0 + \tau_1 \geq 1$, and has positive-length intersection iff $\tau_0 + \tau_1 > 1$.*

2. *$R_{00}$ has nonempty intersection with $\mathcal{M}$ iff $\tau_0 + \tau_1 \leq 1$, and has positive-length intersection iff $\tau_0 + \tau_1 < 1$.*

3. *On the boundary $\tau_0 + \tau_1 = 1$, both $R_{11}$ and $R_{00}$ intersect $\mathcal{M}$ at a single point; hence under any continuous distribution on $\mathcal{M}$ they have probability zero and only the singleton regions carry mass.*

*Proof.* Parameterize $\mathcal{M}$ by $u = s_0 \in [0, 1]$, so $s_1 = 1 - u$.

$R_{11}$ requires $u \leq \tau_0$ and $1 - u \leq \tau_1$, i.e. $u \in [1 - \tau_1,\ \tau_0]$. This interval has positive length iff $1 - \tau_1 < \tau_0$, i.e. $\tau_0 + \tau_1 > 1$, and degenerates to a point at equality.

$R_{00}$ requires $u > \tau_0$ and $1 - u > \tau_1$, i.e. $u \in (\tau_0,\ 1 - \tau_1)$. This interval has positive length iff $\tau_0 < 1 - \tau_1$, i.e. $\tau_0 + \tau_1 < 1$, and degenerates to a point at equality. $\qquad \square$

Crossing the affine boundary $\tau_0 + \tau_1 = 1$ therefore removes an entire region label (doublet or abstention) from the support of $R_\tau(X)$ under probability normalization, explaining sharp regime changes in attainable operating behavior.

### E.3 Cross-threshold dominance in the hedging regime

Within the hedging regime $\tau_0 + \tau_1 > 1$, the manifold $\mathcal{M}$ is partitioned into three contiguous intervals corresponding to $\{10, 11, 01\}$. With $u = s(x, 0)$:

$$u \in [0,\, 1 - \tau_1) \Rightarrow R_\tau(x) = 10, \qquad u \in [\,1 - \tau_1,\, \tau_0\,] \Rightarrow R_\tau(x) = 11, \qquad u \in (\tau_0,\, 1] \Rightarrow R_\tau(x) = 01.$$

Hence each singleton region is controlled primarily by the *opposing* threshold: increasing $\tau_1$ expands 11 at the expense of 10, while increasing $\tau_0$ expands 11 at the expense of 01. Operationally, $(\tau_0, \tau_1)$ act as *mass-reallocation boundaries*, not independent per-class knobs.

Note that under probability normalization, setting thresholds to $(1 - \tau_1, 1 - \tau_0)$ exchanges the roles of regions 11 and 00. If the deployment policy treats hedging and abstention identically, then this swap does not change action-level behavior, even though marginal coverage changes.

### E.4 Threshold classifiers from split conformal cutpoints

When $\tau_0 = \tau_1 = 1/2$, the induced classification rule is the argmax rule (with random tie-breaking), and coverage coincides with the classifier's accuracy.

Using probability-normalized scores $s(x, y) = 1 - P(Y = y \mid X = x)$, the threshold condition $s(x, y) \leq 1/2$ is equivalent to $P(Y = y \mid X = x) \geq 1/2$. Thus, with $\tau_0 = \tau_1 = 1/2$, the conformal set includes exactly those labels whose posterior probability is at least $1/2$: it returns $\{0\}$ when $P(0 \mid x) > 1/2$, $\{1\}$ when $P(1 \mid x) > 1/2$, and (only on ties) $\{0, 1\}$ when $P(0 \mid x) = P(1 \mid x) = 1/2$. If we break ties at random to obtain a single-label predictor, this is precisely the argmax rule. Moreover, away from the tie set the conformal output is a singleton, so the event $\{Y \in \mathcal{C}(X)\}$ is identical to $\{Y = \hat{Y}(X)\}$; hence the coverage probability equals the classification accuracy (with the same tie-breaking convention).

**The $\tau_0 + \tau_1 = 1$ family and asymmetric Bayes costs.** More generally, if $\tau_0 + \tau_1 = 1$ then abstention disappears (up to the tie set) and the conformal output is almost surely a singleton. Indeed, $\mathcal{C}(x)$ includes label $y$ iff $P(Y = y \mid X = x) \geq 1 - \tau_y$, so $\tau_0 + \tau_1 = 1$ implies $1 - \tau_0 = \tau_1$ and $1 - \tau_1 = \tau_0$, yielding the one-parameter threshold rule

$$\hat{Y}(x) = \mathbf{1}\{P(1 \mid x) \geq \tau_0\} = \mathbf{1}\{P(0 \mid x) \leq \tau_1\}.$$

In this regime, $\Pr(Y \in \mathcal{C}(X)) = \Pr(Y = \hat{Y}(X))$, i.e., coverage equals the accuracy of the corresponding threshold classifier.

The parameter $\tau_0$ also has a standard decision-theoretic interpretation. Consider asymmetric misclassification costs, with $c_{01}$ the cost of a false positive (predicting 1 when $Y = 0$) and $c_{10}$ the cost of a false negative (predicting 0 when $Y = 1$). The Bayes rule that minimizes conditional expected cost predicts 1 whenever

$$c_{01}\, P(Y = 0 \mid x) \;\leq\; c_{10}\, P(Y = 1 \mid x) \qquad \Longleftrightarrow \qquad P(1 \mid x) \;\geq\; \frac{c_{01}}{c_{01} + c_{10}}.$$

Thus sweeping $\tau_0$ over $(0, 1)$ traces the familiar family of cost-sensitive Bayes classifiers, with $\tau_0 = c_{01}/(c_{01} + c_{10})$ corresponding to the cost ratio $c_{01} : c_{10}$.

### E.5 Region–label primitives and rate factorizations

For auditing and planning, the primitive object is the region–class label table

$$p_{r,y}(\tau) := \Pr(R_\tau(X) = r,\ Y = y \mid \mathcal{D}_{\text{cal}}), \qquad r \in \{00, 01, 10, 11\},\ y \in \{0, 1\}.$$

Two derived summaries are

$$\mu_r(\tau) := \Pr(R_\tau(X) = r \mid \mathcal{D}_{\text{cal}}) = \sum_{y \in \{0,1\}} p_{r,y}(\tau),$$

$$\eta_r(\tau) := \Pr(Y = 1 \mid R_\tau(X) = r,\ \mathcal{D}_{\text{cal}}) = \frac{p_{r,1}(\tau)}{\mu_r(\tau)} \quad (\mu_r(\tau) > 0).$$

Any region-associated KPI is a projection of $\{p_{r,y}(\tau)\}$; see Section 2.4.

**Example: decisive error masses under commit-on-singletons.** Consider the convention

$$10 \mapsto 0, \qquad 01 \mapsto 1, \qquad 11, 00 \mapsto \text{defer / reject.}$$

Then the decisive false-negative and false-positive masses are

$$\text{FN}_{\text{dec}}(\tau) = \Pr(R_\tau(X) = 10, \ Y = 1 \mid \mathcal{D}_{\text{cal}}) = p_{10,1}(\tau) = \mu_{10}(\tau) \, \eta_{10}(\tau),$$

$$\text{FP}_{\text{dec}}(\tau) = \Pr(R_\tau(X) = 01, \ Y = 0 \mid \mathcal{D}_{\text{cal}}) = p_{01,0}(\tau) = \mu_{01}(\tau) \, [1 - \eta_{01}(\tau)].$$

The decisive mass is $\mu_{10}(\tau) + \mu_{01}(\tau)$, while the defer mass is $\mu_{11}(\tau) + \mu_{00}(\tau)$ (with feasibility of 11 versus 00 governed by Proposition 2 under probability normalization).

## F Conformal interface-relative decision optimality and inverse pricing envelopes for a fixed conformal interface

This appendix formalizes the operational point that distribution-free calibration is not automatically cost-agnostic once conformal outputs are wired into actions. We specialize classical statistical decision theory and reject-option analysis (Casella & Berger, 2002; Chow, 1970; Herbei & Wegkamp, 2006; Yuan & Wegkamp, 2010; Bartlett & Wegkamp, 2008) to the conformal interface as the deployed observable. Fix thresholds $\tau$ (calibration-conditional viewpoint) and treat the induced finite region label $R_\tau(X) \in \mathcal{R}$ as the deployed observable. Any downstream rule that uses only this interface can depend on the data only through $r \in \mathcal{R}$, so whether the convention is decision-optimal relative to the conformal interface under a given cost model is evaluated *region-wise* using within-region label frequencies (we formalize this below as *conformal interface-relative decision optimality*). The underlying concept is simple: among actions available after observing only $R_\tau(X)$, the deployed convention should minimize region-wise conditional expected loss. We cast this as an *inverse pricing* problem: given a fixed (interface, convention) pair, characterize the set of consequence prices under which the convention is decision-optimal relative to the conformal interface.

The main point is clear: once a conformal predictor is treated as a decision interface, decision-optimal downstream action depends on the region-wise label frequencies, not on coverage alone and not on the set-valued output structure alone. The contribution of this appendix is not new decision theory; it is to make that dependence explicit for a fixed conformal interface through a worked Chow-style case.

### F.1 Interface primitives: masses and within-region label frequencies

Specialize to $\mathcal{Y} = \{0, 1\}$ and $\mathcal{R} = \{00, 01, 10, 11\}$ as in Appendix E. Adopt the calibration-conditional joint table

$$p_{r,y} := \mathbb{P}(R_\tau(X) = r, \ Y = y \mid \mathcal{D}_{\text{cal}}).$$

Define region mass and within-region label frequency

$$\mu_r := p_{r,0} + p_{r,1}, \qquad \eta_r := \Pr(Y = 1 \mid R_\tau(X) = r, \ \mathcal{D}_{\text{cal}}) = \frac{p_{r,1}}{\mu_r} \quad (\mu_r > 0).$$

Because $\mathcal{R}$ is finite, conditional uncertainty about $Y$ is piecewise constant: within region $r$, all decision comparisons reduce to $\eta_r$.

### F.2 Region-associated action conventions and decision optimality

Let $\mathcal{A}$ be a finite action set and let $L_\theta(a, y)$ be a priced consequence (cost or negative utility), parameterized by $\theta \in \Theta$. A deployed convention is a region-associated policy

$$\tilde{\pi} : \mathcal{R} \to \mathcal{A}.$$

For region $r$ with $\mu_r > 0$, the interface-relative conditional risk of action $a$ is

$$\mathcal{C}_\theta(a \mid r) := \mathbb{E}[L_\theta(a, Y) \mid R_\tau(X) = r, \ \mathcal{D}_{\text{cal}}] = (1 - \eta_r) L_\theta(a, 0) + \eta_r L_\theta(a, 1).$$

**Conformal interface-relative decision optimality.** We say $\tilde{\pi}$ is *decision-optimal relative to the conformal interface* under pricing $\theta$ if for every region with $\mu_r > 0$,

$$\mathcal{C}_\theta(\tilde{\pi}(r) \mid r) \leq \mathcal{C}_\theta(a \mid r) \qquad \forall a \in \mathcal{A}.$$

This is precisely decision optimality relative to the coarsened observable $R_\tau(X)$: the action chosen in each region minimizes posterior expected loss among the admissible actions available at that interface.

### F.3 Inverse pricing envelope

For each region with $\mu_r > 0$, define the local feasibility set

$$\Theta_r(\tilde{\pi}) := \Big\{ \theta \in \Theta : \ \mathcal{C}_\theta(\tilde{\pi}(r) \mid r) \leq \mathcal{C}_\theta(a \mid r) \ \ \forall a \in \mathcal{A} \Big\},$$

and the global pricing envelope

$$\Theta(\tilde{\pi}) := \bigcap_{r:\mu_r>0} \Theta_r(\tilde{\pi}).$$

Equivalently, $\Theta(\tilde{\pi})$ is the set of price parameters for which the forced action in each region is decision-optimal relative to the conformal interface given only the conformal output. Since only action comparisons matter, envelopes are naturally reported in ratio (projective) coordinates: global positive scaling of $L_\theta$ is irrelevant, and adding offsets $b_y$ independent of $a$ preserves comparisons.

### F.4 Worked case: Chow-style reject option and commit on singletons

Consider $\mathcal{A} = \{0, 1, \text{rej}\}$ with Chow-style costs (false negative $c_{01} > 0$, false positive $c_{10} > 0$, rejection $c_{\text{rej}} \geq 0$):

$$L(0,1) = c_{01}, \quad L(1,0) = c_{10}, \quad L(\text{rej}, y) = c_{\text{rej}}, \quad L(0,0) = L(1,1) = 0.$$

In region $r$ the conditional risks are

$$\mathcal{C}(0 \mid r) = \eta_r\, c_{01}, \qquad \mathcal{C}(1 \mid r) = (1 - \eta_r)\, c_{10}, \qquad \mathcal{C}(\text{rej} \mid r) = c_{\text{rej}}.$$

Work in ratios

$$\lambda := \frac{c_{01}}{c_{10}}, \qquad \rho := \frac{c_{\text{rej}}}{c_{10}},$$

so only $(\lambda, \rho)$ matters for comparisons.

**Convention.**

$$\tilde{\pi}(10) = 0, \qquad \tilde{\pi}(01) = 1, \qquad \tilde{\pi}(11) = \text{rej}, \qquad \tilde{\pi}(00) = \text{rej}.$$

This convention is decision-optimal relative to the conformal interface iff the region-wise dominance inequalities below hold for all regions with $\mu_r > 0$.

**Singleton region** $01$ **(output $\{1\}$).** Choosing 1 must beat both 0 and rej:

$$(1 - \eta_{01}) \leq \eta_{01}\lambda \quad \Longleftrightarrow \quad \eta_{01} \geq \frac{1}{1 + \lambda}, \qquad (1 - \eta_{01}) \leq \rho \quad \Longleftrightarrow \quad \eta_{01} \geq 1 - \rho.$$

Thus

$$\eta_{01} \ \geq \ \max\Big\{ \frac{1}{1 + \lambda}, \ 1 - \rho \Big\}.$$

**Singleton region** $10$ **(output $\{0\}$).** Choosing 0 must beat both 1 and rej:

$$\eta_{10}\lambda \leq (1 - \eta_{10}) \quad \Longleftrightarrow \quad \eta_{10} \leq \frac{1}{1 + \lambda}, \qquad \eta_{10}\lambda \leq \rho \quad \Longleftrightarrow \quad \eta_{10} \leq \frac{\rho}{\lambda}.$$

Thus

$$\eta_{10} \ \leq \ \min\Big\{ \frac{1}{1 + \lambda}, \ \frac{\rho}{\lambda} \Big\}.$$

**Rejection regions $r \in \{11, 00\}$.** Rejecting must beat both commitments:

$$\rho \leq \eta_r \lambda \quad \Longleftrightarrow \quad \eta_r \geq \frac{\rho}{\lambda}, \qquad \rho \leq (1 - \eta_r) \quad \Longleftrightarrow \quad \eta_r \leq 1 - \rho.$$

Hence rejection is optimal on region $r$ only if

$$\frac{\rho}{\lambda} \ \leq \ \eta_r \ \leq \ 1 - \rho.$$

The rejection band is nonempty only if

$$\frac{\rho}{\lambda} \leq 1 - \rho \quad \Longleftrightarrow \quad \rho \leq \frac{\lambda}{1 + \lambda} \quad \Longleftrightarrow \quad c_{\text{rej}} \leq \frac{c_{01} c_{10}}{c_{01} + c_{10}}.$$

**Polyhedral summary.** The conditions above can be summarized as a pricing envelope in $(\lambda, \rho)$-space. Because they are linear inequalities, the admissible cost set is a convex polytope (intersection of half-spaces), as is standard in reject-option decision theory (Yuan & Wegkamp, 2010). For the convention $\tilde{\pi}(10) = 0$, $\tilde{\pi}(01) = 1$, $\tilde{\pi}(11) = \tilde{\pi}(00) = \text{rej}$, the envelope is the set of $(\lambda, \rho) \in \mathbb{R}^2_{>0}$ satisfying

$$\lambda \geq \frac{1 - \eta_{01}}{\eta_{01}} \quad \text{[singleton 01: commit to 1 beats 0]}, \tag{3}$$

$$\lambda \leq \frac{1 - \eta_{10}}{\eta_{10}} \quad \text{[singleton 10: commit to 0 beats 1]}, \tag{4}$$

$$\rho \geq 1 - \eta_{01} \quad \text{[singleton 01: commit beats reject]}, \tag{5}$$

$$\rho \geq \lambda \eta_{10} \quad \text{[singleton 10: commit beats reject]}, \tag{6}$$

$$\rho \leq \lambda \eta_r, \quad \rho \leq 1 - \eta_r \quad \text{[rejection regions } r \in \{11, 00\} \text{ with } \mu_r > 0], \tag{7}$$

together with $\lambda > 0$ and $\rho \geq 0$.

### F.5 Follow-up observations

**Monotone ordering.** A useful corollary of the inequalities is that the pricing envelope is non-empty only when the rejection regions have intermediate label composition: $\eta_{10} \leq \eta_{01}$ and, for each populated $r \in \{11, 00\}$, $\eta_{10} \leq \eta_r \leq \eta_{01}$. This is the region-wise analogue of the classical reject-option condition from Chow (1970); Herbei & Wegkamp (2006): rejection is decision-optimal only when the posterior falls in an intermediate band.

**Coverage-matched settings.** Holding the scoring model and deployment distribution fixed, different calibration settings $\tau$ and $\tau'$ can have the same marginal coverage but different region compositions, and therefore different pricing envelopes. In some cases those envelopes may even fail to overlap. The point is not tied to different datasets; it is a statement about how the same underlying interface can behave under different calibration choices. This is the conformal-specific implication of the preceding decision-theoretic setup: coverage-matched rules may still be decision-optimal under different parts of cost-ratio space. Section 5.3 and Figure 5 show the milder empirical version of the same idea, namely that the feasible $(\lambda, \rho)$ region varies across Pareto regimes.

**Dual view.** The same inequalities can be read in reverse: given $(\lambda, \rho)$, which action convention is decision-optimal relative to the conformal interface? This partitions cost-ratio space into regions where different conventions are decision-optimal—a standard dual perspective in cost-sensitive classification (Bartlett & Wegkamp, 2008; Yuan & Wegkamp, 2010). For the present appendix, this is best read as an interpretive lens on the same region-wise quantities rather than as a separate result.

**Summary.** The main point of this appendix is the same one stated at the start: for a fixed calibrated conformal predictor and a fixed wiring convention $\tilde{\pi}$, decision optimality relative to the conformal interface depends on the region-wise label frequencies through the posteriors $\{\eta_r\}$. Coverage alone is not enough,

and the bare prediction-set structure is not enough either. The worked Chow-style case, together with the follow-up observations on monotone ordering (F.5) and coverage-matched settings (F.5), simply makes that dependence explicit for the conformal interface and shows how the region–class label table restricts which cost ratios make a given downstream convention decision-optimal.

## G   Explicit region indicators and projection masks

This appendix briefly instantiates the linear "sum selected cells" formalism from Section 2.4 in the binary geometry used in Figure 2 and the policy-projection example from the main text $(R_\tau(x) \xrightarrow{\pi} C(x))$. Region feasibility and regime facts are recorded in Appendix E; here we keep only the explicit projection masks needed for audit computations.

### G.1   Region labels and the region–class label table

Assume $\mathcal{Y} = \{0,1\}$ and thresholds $\tau = (\tau_0, \tau_1)$. Define the deployed region label

$$R_\tau(x) = (r_0(x), r_1(x)) \in \{0,1\}^2, \qquad r_y(x) := \mathbf{1}\{s(x,y) \le \tau_y\},$$

with region names $\mathcal{R} = \{r_{10}, r_{11}, r_{01}, r_{00}\}$ as in Appendix E.1. For a calibration setting $\theta$ (indexing deployed thresholds $\tau(\theta)$), define the calibration-conditional region–class label table

$$P(\theta) = \big(p_{r,y}(\theta)\big)_{r \in \mathcal{R},\ y \in \mathcal{Y}}, \qquad p_{r,y}(\theta) = \Pr\big(R_{\tau(\theta)}(X) = r,\ Y = y \mid \mathcal{D}_{\text{cal}}\big).$$

A linear operational rate is obtained by summing the subset of cells $(r,y)$ that define the event.

### G.2   Policies used in Figure 3

A policy $\pi$ maps region labels to prediction sets $C(x) \subseteq \{0,1\}$:

$$R_\tau(x) \xrightarrow{\pi} C(x).$$

The three policies used in Figure 3 are:

**Set inclusion $\pi_{\text{SI}}$.**

$$\pi_{\text{SI}}(r_{10}) = \{0\}, \quad \pi_{\text{SI}}(r_{11}) = \{0,1\}, \quad \pi_{\text{SI}}(r_{01}) = \{1\}, \quad \pi_{\text{SI}}(r_{00}) = \varnothing.$$

**Commit–reject $\pi_{\text{CR}}$.**

$$\pi_{\text{CR}}(r_{10}) = \{0\}, \quad \pi_{\text{CR}}(r_{01}) = \{1\}, \quad \pi_{\text{CR}}(r_{11}) = \varnothing, \quad \pi_{\text{CR}}(r_{00}) = \varnothing.$$

**Set exclusion $\pi_{\text{SE}}$ (complement of set inclusion).**

$$\pi_{\text{SE}}(r_{10}) = \{1\}, \quad \pi_{\text{SE}}(r_{11}) = \varnothing, \quad \pi_{\text{SE}}(r_{01}) = \{0\}, \quad \pi_{\text{SE}}(r_{00}) = \{0,1\}.$$

### G.3   Binary projection masks

Fix a policy $\pi$. For any event/quantity $\ell$, define a $4 \times 2$ binary mask

$$G_\ell(\pi) = \big(g_\ell(r,y;\pi)\big)_{r \in \mathcal{R},\ y \in \mathcal{Y}} \in \{0,1\}^{4 \times 2},$$

where $g_\ell(r,y;\pi) = 1$ indicates that the cell $(r,y)$ is included in the sum. Then the corresponding rate is

$$r_\ell(\theta) = \sum_{r \in \mathcal{R}} \sum_{y \in \mathcal{Y}} g_\ell(r,y;\pi)\, p_{r,y}(\theta).$$

We record one detailed worked example and then list several shorter special cases used elsewhere in the paper.

**(1) Coverage under set inclusion.** Under $\pi_{\mathrm{SI}}$, coverage is the event $\{Y \in C(X)\}$. Cell-by-cell, coverage holds for: (i) $y = 0$ in regions $r_{10}$ or $r_{11}$, and (ii) $y = 1$ in regions $r_{01}$ or $r_{11}$. Hence

$$
G_{\mathrm{cov}}(\pi_{\mathrm{SI}}) =
\begin{array}{c|cc}
 & y = 0 & y = 1 \\
\hline
r_{10} & 1 & 0 \\
r_{11} & 1 & 1 \\
r_{01} & 0 & 1 \\
r_{00} & 0 & 0
\end{array}
$$

and therefore

$$
r_{\mathrm{cov}}(\theta) = p_{10,0}(\theta) + p_{11,0}(\theta) + p_{11,1}(\theta) + p_{01,1}(\theta).
$$

Equivalently, coverage fails on the two singleton mistakes $(r_{10}, 1)$ and $(r_{01}, 0)$ and on all abstentions $r_{00}$.

**(2) Conformal Efficiency** In order to characterize the amount of uncertainty carried by a conformal predictor, the concept of conformal efficiency (Heyndrickx et al., 2023) has been proposed. Conformal efficiency is essentially defined as the total singleton rate. The region mask for conformal efficiency is obtained by summing the cells that define the event:

$$
G_{\mathrm{CE}}(\pi_{\mathrm{SI}}) =
\begin{array}{c|cc}
 & y = 0 & y = 1 \\
\hline
r_{10} & 1 & 1 \\
r_{11} & 0 & 0 \\
r_{01} & 1 & 1 \\
r_{00} & 0 & 0
\end{array}
$$

and therefore

$$
r_{\mathrm{CE}}(\theta) = p_{10,0}(\theta) + p_{10,1}(\theta) + p_{01,0}(\theta) + p_{01,1}(\theta).
$$

Under $\pi_{\mathrm{SI}}$, coverage holds automatically on hedged outputs ($r_{11}$), while coverage failures occur only on singleton mistakes and abstentions. In particular,

$$
r_{\mathrm{cov}}(\theta) = p_{10,0}(\theta) + p_{11,0}(\theta) + p_{11,1}(\theta) + p_{01,1}(\theta),
$$

so that

$$
r_{\mathrm{cov}}(\theta) - r_{\mathrm{CE}}(\theta) = \big(p_{11,0}(\theta) + p_{11,1}(\theta)\big) - \big(p_{10,1}(\theta) + p_{01,0}(\theta)\big).
$$

Thus, when calibration places substantial mass in the hedging region $r_{11}$, coverage can exceed conformal efficiency because hedged prediction sets are always covered. If the calibration is performed in the abstention regime ($r_{00}$), then coverage and conformal efficiency differ by the total error mass among singleton predictions.

**(3) Other projections used in the paper.** The same mask formalism gives the remaining quantities used in Figure 1 and in the binary operational examples:

$$
\text{missed positive mass:} \quad q_{10}(\theta) = \Pr(Y = 1,\ C(X) = \{0\}) = \Pr(Y = 1,\ R_\tau(X) = r_{10}) = p_{10,1}(\theta),
$$

$$
\text{hedged positive mass:} \quad q_{11}(\theta) = \Pr(Y = 1,\ C(X) = \{0, 1\}) = \Pr(Y = 1,\ R_\tau(X) = r_{11}) = p_{11,1}(\theta),
$$

$$
\text{abstention mass under } \pi_{\mathrm{CR}}: \quad r_{\mathrm{abs}}(\theta; \pi_{\mathrm{CR}}) = \big(p_{11,0}(\theta) + p_{11,1}(\theta)\big) + \big(p_{00,0}(\theta) + p_{00,1}(\theta)\big).
$$

These correspond respectively to a one-cell projection, another one-cell projection, and a two-row projection. Their binary masks are obtained immediately by placing ones on the selected cells.

### G.4 Ratios of projections (conditional diagnostics)

Some diagnostics are conditional probabilities and therefore ratios of linear sums. For example, the purity of singleton-1 outputs under $\pi_{\mathrm{SI}}$ is

$$
\mathrm{Purity}_1(\theta) := \Pr\big(Y = 1 \mid C(X) = \{1\}\big) = \frac{\Pr(Y = 1,\ C(X) = \{1\})}{\Pr(C(X) = \{1\})} = \frac{p_{01,1}(\theta)}{p_{01,0}(\theta) + p_{01,1}(\theta)}.
$$

Table 4: Tox21 dataset composition and effective average class-conditional calibration sizes under the experimental protocol.

| Endpoint | Total Samples | Positive Rate | Calib. Positives | Calib. Negatives |
|---|---|---|---|---|
| NR-AR | 7265 | 4.3% | 77 | 1739 |
| NR-AR-LBD | 6758 | 3.5% | 59 | 1630 |
| NR-AhR | 6549 | 11.7% | 192 | 1445 |
| NR-Aromatase | 5821 | 5.2% | 75 | 1380 |
| NR-ER | 6193 | 12.8% | 198 | 1350 |
| NR-ER-LBD | 6955 | 5.0% | 87 | 1651 |
| NR-PPAR-$\gamma$ | 6450 | 2.9% | 46 | 1566 |
| SR-ARE | 5832 | 16.2% | 235 | 1223 |
| SR-ATAD5 | 7072 | 3.7% | 66 | 1702 |
| SR-HSE | 6467 | 5.8% | 93 | 1523 |
| SR-MMP | 5810 | 15.8% | 229 | 1223 |
| SR-p53 | 6774 | 6.2% | 105 | 1587 |

Table 5: **Aggregated Tox21 coverage and set statistics.** Results are averaged over twelve endpoints and 100 random splits.

| Method | Mean Coverage | Violation Rate | Avg. Set Size | Singleton Rate |
|---|---|---|---|---|
| Standard Split Conformal | 0.917 | 0.305 | 1.41 | 0.52 |
| DKWM Correction | 0.986 | 0.005 | 1.78 | 0.22 |
| SSBC | 0.951 | 0.068 | 1.54 | 0.40 |

**Audit computation.** All entries of $P(\theta)$ are estimated from region–class label counts on the audit set. Linear KPIs are computed by summing selected cells; conditional diagnostics are computed as ratios of such sums.

## H   Tox21 supplementary details

This appendix provides supplementary documentation for the Tox21 experiments reported in Section 5.2. The purpose of this appendix is reproducibility and contextualization rather than extension of results or additional claims.

### H.1   Tox21 dataset

The Tox21 benchmark consists of binary toxicity outcomes for twelve biological assays spanning nuclear receptor (NR) signaling and cellular stress response (SR) pathways. Each compound is labeled as *active* or *inactive* per assay. Labels are sparse and highly imbalanced, particularly for nuclear receptor targets.

The dataset composition table 4 and aggregated coverage summary 5 are recorded here.

### H.2   Representative endpoint-level operational summaries

The main text reports SR-MMP as a representative moderate-prevalence endpoint. We keep NR-AR here to document a complementary low-prevalence regime.

Because conformal calibration is performed in a class-conditional (Mondrian) manner, the effective calibration size for the positive class governs feasibility and discretization effects.

### H.3   Representation and model protocol

All Tox21 experiments use a fixed, task-agnostic molecular representation (descriptors plus Morgan fingerprints) and a fixed CatBoost training protocol (1000 iterations, depth 6, learning rate 0.1, log-loss), with no assay-specific feature engineering, class reweighting, or resampling. Molecules are parsed and sanitized using

Table 6: **NR-AR endpoint.** Operational performance summary for the NR-AR endpoint. All reported quantities are joint probabilities normalized by the total test-set size. Rows report the singleton rate, doublet rate, and wrong-singleton rate ($P(Y = c, |S| = 1, \hat{y} \neq Y)$) by true class. "$\mathcal{D}_1$" and "LOO 95% PI $\mathcal{D}_1$" denote leave-one-out point estimates and their beta–binomial planning intervals computed on the calibration data. "$\mathcal{D}_2$" reports empirical rates on the audit set. "BB 95% PI $\mathcal{D}_2$" gives Beta–Binomial predictive summaries for the corresponding $\mathcal{D}_2$ quantities over a future window of the same size.

| Operational quantity | Class | $\mathcal{D}_1$ | LOO 95% PI $\mathcal{D}_1$ | $\mathcal{D}_2$ | BB 95% PI $\mathcal{D}_2$ |
|---|---|---|---|---|---|
| Singleton rate | Class 0 | 0.163 | $[0.125, 0.205]$ | 0.133 | $[0.112, 0.157]$ |
| | Class 1 | 0.026 | $[0.011, 0.047]$ | 0.029 | $[0.019, 0.041]$ |
| Doublet rate | Class 0 | 0.796 | $[0.751, 0.838]$ | 0.818 | $[0.792, 0.843]$ |
| | Class 1 | 0.015 | $[0.005, 0.031]$ | 0.020 | $[0.012, 0.030]$ |
| Wrong-singleton rate | Class 0 | 0.086 | $[0.058, 0.119]$ | 0.060 | $[0.045, 0.077]$ |
| | Class 1 | 0.002 | $[0.000, 0.010]$ | 0.001 | $[0.000, 0.004]$ |

standard cheminformatics tooling, and compounds with invalid features are excluded. This intentionally non-optimized setup isolates conformal calibration and operational-envelope behavior from model-architecture tuning; observed AUROC values range between 0.80–0.85 across endpoints, and are consistent with baseline Tox21 classifiers.

### H.4 Conformal calibration and evaluation

All conformal predictors are calibrated in a class-conditional (Mondrian) fashion, with separate calibration sets for the positive and negative classes. For each run, one of three calibration strategies is applied: standard split conformal, DKWM-based correction, or SSBC, targeting $(\alpha, \delta) = (0.10, 0.10)$.

Calibration thresholds are computed once per run and evaluated on an independent held-out split. In the main text tables, the " $\mathcal{D}_2$" columns report the corresponding endpoint-level operational rates on that held-out split, which serve as the audit-based reference for the operational claims. The "$\mathcal{D}_1$" and "LOO 95% PI $\mathcal{D}_1$" columns are single-sample planning summaries computed from leave-one-out recalibration on the calibration split. They are included to compare the LOO planning proxy to the independent audit reference.

## I Solubility supplementary details

This appendix records methodological details for the solubility scenario-planning experiments in Section 5.3. We first describe the dataset, partitioning, modeling, and calibration setup, and then document the operational artifacts used to interpret the resulting Pareto front.

### I.1 Dataset and label construction

We use AquaSolDB as the source of aqueous solubility measurements ($\log S$), following the curation and quality-control procedures described in the dataset reference (Sorkun et al., 2019). The curated source file used in our experiments contains 9,982 entries. Molecules with invalid SMILES strings are excluded before feature generation. For model training and scoring we discretize $\log S$ into three regimes with fixed thresholds: Insoluble ($\log S < -4$), Moderate ($-4 \leq \log S < -2$), and Soluble ($\log S \geq -2$), defining a three-class label space $\mathcal{Y} = \{0, 1, 2\}$ (Kalepu & Nekkanti, 2015).

### I.2 Partitioning: scaffold-based training and stratified calibration/test split

We use a hybrid split to separate scaffold-aware model fitting from the calibration pool used in the LOO planning analysis:

- **Scaffold-based training split.** Molecules are grouped by Bemis–Murcko scaffold. The scaffold groups are shuffled with a fixed seed, and groups are added to the training split until roughly 70% of molecules are assigned. This makes the training split scaffold-aware rather than purely random.

- **Stratified calibration/test split.** The remaining molecules are split equally between calibration and test using stratified random sampling over the three-class target.

- **Calibration/test split for diagnostics.** The calibration/test split is retained for exploratory diagnostics and documentation of the scenario construction. The reported planning quantities in Section 5.3 are not obtained from an independent audit/test evaluation; they are obtained from the leave-one-out planning interface on the scenario-restricted calibration sample.

### I.3   Molecular representation

Molecules are represented using concatenated multi-resolution Morgan fingerprints (Rogers & Hahn, 2010) computed with RDKit. We use three fingerprint tiers: radius 4 with 512 bits, radius 3 with 1024 bits, and radius 2 with 2048 bits, yielding 3584 binary features per molecule. Molecules that fail RDKit parsing are excluded before feature generation.

### I.4   Model and training protocol

We train a CatBoost (Prokhorenkova et al., 2018) gradient-boosted decision tree classifier on the scaffold-based training split with fixed hyperparameters: 1000 boosting iterations, depth 6, learning rate 0.05, random seed 42, and multi-class loss. After training, the predictor is frozen and treated as infrastructure; the experiments study the conformal and operational layers rather than optimizing base accuracy.

### I.5   Calibration and operationalization

Uncertainty quantification uses split conformal prediction with SSBC calibration (Section 5.1.1 and Appendix C), which selects a grid index to stabilize realized coverage semantics at user-specified confidence (Vovk, 2012a). In the solubility case study, all reported planning quantities are implemented through the leave-one-out (LOO) construction described in Appendix D, rather than through an independent audit split. This calibration layer therefore supports the finite-window planning-envelope constructions used for operational planning. The calibration/test split is retained as supporting context for the scenario construction, but the reported trade-off maps and planning quantities are generated from the LOO interface on the scenario-restricted calibration sample.

Although the classifier is trained on three classes (Insoluble, Moderate, Soluble), the planning analysis in Section 5.3 uses a binary operational objective obtained by merging Moderate and Soluble into a single Soluble class. Conformal prediction sets are computed in the binary label space {Insol, Sol} and operational quantities (loss, waste, hedging, decisiveness) are computed with respect to this binary interface.

### I.6   Deployment-matched calibration via chemical tribes

Scenario planning conditions exchangeability on a deployment-defining event by restricting calibration to a chemically defined subpopulation.

**Tribe construction.**   We define coarse chemical "tribes" using RDKit MolLogP:

$$\texttt{Tribe\_Lipophilic} : \text{MolLogP} > 3.5,$$
$$\texttt{Tribe\_Hydrophilic} : \text{MolLogP} < 1.0,$$
$$\texttt{Tribe\_Neutral} : 1.0 \leq \text{MolLogP} \leq 3.5.$$

We compute summary statistics of nonconformity and class probabilities stratified by tribe to document and motivate the focal deployment regime used in the planning study.

**Focal deployment scenario.** Section 5.3 focuses on a lipophilic deployment regime, targeting molecules that are not naturally hydrophilic. SSBC calibration is therefore performed on the restricted sample of lipophilic molecules:

$$\mathcal{D}_{\text{cal}}^{\text{lip}} = \{(X_i, Y_i) \in \mathcal{D}_{\text{cal}} : \text{MolLogP}(X_i) > 3.5\},$$

treated as exchangeable with respect to the intended deployment distribution. The cutoff $\text{MolLogP} > 3.5$ is used here as a domain-motivated scenario definition for lipophilic compounds, not as a tuned parameter chosen to optimize the observed trade-off map. After down-selection to this scenario and merging Moderate and Soluble into the binary soluble label, the resulting planning dataset contains 319 entries: 53 soluble and 266 insoluble. These counts describe the scenario dataset over which the planning sweep is performed; they do not correspond to a single selected operating point.

**Scenario-conditional exchangeability assumption.** This restriction is scenario conditioning, not a general covariate-shift correction. We assume deployment draws satisfy the same scenario event $E = \{\text{MolLogP}(X) > 3.5\}$ so calibration and deployment are exchangeable conditional on $E$. If deployment violates $E$, or if exchangeability fails within $E$, conformal validity is not guaranteed. Shift-robust validity methods are complementary and outside scope; see, e.g., Fannjiang et al. (2022).

**Interpretation.** Conditioning trades calibration sample size for improved deployment match; the resulting loss of statistical efficiency appears directly as stronger SSBC corrections and wider planning envelopes, consistent with transparent planning under finite-sample uncertainty.

## I.7 Operational outcome definitions

Operational outcomes are defined in the main text via the binary prediction-set interface (TP, FN, HP, FP, TN, HN). All reported quantities in the scenario-planning analysis are joint rates over the set–label outcomes $P(Y = y, C = c)$, scaled to expected counts over a window of $m = 1000$ molecules. For this case study, these quantities are estimated using pooled LOO indicators and the associated planning-envelope construction. They are planning summaries for the restricted scenario sample and exhibit the structure of feasible trade-offs under the chosen scenario definition.

## I.8 Additional Pareto-front details

We record the representative Pareto-optimal regimes here and then summarize the appendix-specific details behind the solubility Pareto sweep, in particular how the nominal SSBC parameters function along the front and how to interpret $\alpha$ versus $\delta$ once the scenario-restricted planning interface has been fixed.

## I.9 Functional roles of calibration parameters

The Pareto sweep in Section 5.3 varies the four nominal parameters $(\alpha_0, \delta_0, \alpha_1, \delta_1)$ (with SSBC mapping them to effective deployed grid levels). Empirically the knobs are not interchangeable: changes reallocate mass among outcome categories (loss, waste, hedging, decisiveness) along channels constrained by the underlying threshold geometry.

Throughout, class 0 denotes insoluble and class 1 denotes soluble. The qualitative roles below summarize matched Pareto-optimal solutions where one parameter varies while the others are held fixed; rates are computed from the provided Pareto-front CSV and reported as events per 1000 molecules.

**Role of $\alpha_0$: global conservatism against irreversible loss.** Decreasing $\alpha_0$ suppresses loss by expanding hedging; increasing $\alpha_0$ collapses ambiguity into more decisive singleton predictions. In many matched segments, the dominant effect is redistribution between hedged and singleton outcomes with loss already near saturation.

**Role of $\delta_0$: fine-scale sharpening on the insoluble side.** For fixed $(\alpha_0, \alpha_1, \delta_1)$, $\delta_0$ tends to act locally, shifting mass between hedged and singleton insoluble predictions with limited impact on loss.

Table 7: Two representative Pareto-optimal operating regimes for the solubility scenario, illustrating contrasting planning postures. Rates are expected counts per 1000 molecules, with planning envelopes in brackets.

| | Loss-minimizing regime | | High-decisiveness regime | |
|---|---|---|---|---|
| $\alpha_0$ | 0.125 | | 0.150 | |
| $\delta_0$ | 0.150 | | 0.150 | |
| $\alpha_1$ | 0.050 | | 0.150 | |
| $\delta_1$ | 0.075 | | 0.100 | |
| | Rate | Interval | Rate | Interval |
| Loss rate | 3 | $[0, 35]$ | 12 | $[0, 58]$ |
| Waste rate | 70 | $[27, 130]$ | 13 | $[0, 49]$ |
| Total hedge rate | 785 | $[663, 896]$ | 514 | $[376, 675]$ |
| Correct soluble singleton rate | 7 | $[0, 38]$ | 88 | $[40, 155]$ |
| Correct insoluble singleton rate | 135 | $[68, 215]$ | 282 | $[196, 375]$ |
| Decisiveness $(1000 - \text{hedge})$ | 215 | | 486 | |

**Role of $\alpha_1$: boundary-controlled tolerance with nonlocal effects.** Although $\alpha_1$ is nominally associated with the soluble side, changing $\alpha_1$ can move the operating point across geometric boundaries, inducing nonlocal reallocations that may appear most strongly in insoluble outcomes.

**Role of $\delta_1$: systematic hedge-to-insoluble reassignment.** Across stable regions of the front, increasing $\delta_1$ often converts a portion of hedged mass into insoluble singletons, reflecting how the normalized score constraints and threshold geometry position the hedging interval.

**Geometric interpretation.** Overall, the Pareto front is shaped by geometric coupling rather than independent tuning: each knob reallocates probability mass along constrained channels determined by the threshold partition and the finite-sample SSBC adjustment.

### I.10 Interpreting $\alpha$ versus $\delta$ on the Pareto front

Within SSBC the deployed threshold corresponds to an adjusted effective level $\tilde{\alpha}$ determined jointly by $(\alpha, \delta)$. Thus $\alpha$ and $\delta$ are best viewed as two parameterizations of a single underlying control (the effective threshold), with different practical resolution under finite-sample constraints. The approximate feasibility relation $\delta \approx (1 - \alpha)^N$ implies

$$\log \delta = N \log(1 - \alpha),$$

so changes in $\delta$ correspond to fine-grained (approximately logarithmic) adjustments in $\tilde{\alpha}$, whereas changes in $\alpha$ on a coarse grid can move the operating point across qualitatively different regions of the feasible manifold. This helps explain why $\alpha$ sweeps can trigger large reallocations among loss/waste/hedging outcomes, while $\delta$ sweeps more often act as local "sharpening" of decisions within a geometric regime.

