# OpenReview forum: "Conformal Tradeoffs: Operational Profiles Beyond Coverage"
_TMLR — Rejected by TMLR_

### Review · Reviewer_yfDs · 2026-03-22

**Summary Of Contributions:**

The paper studies conformal prediction from a deployment perspective, arguing that coverage alone is insufficient to characterize real-world behavior. It introduces a framework where a calibrated model induces a region-class label table, which serves as a reusable summary to compute operational metrics. To anchor coverage guarantees in finite samples, the authors propose Small-Sample Beta Correction (SSBC), which adjusts conformal calibration to achieve any user-specified PAC-style coverage guarantee. They then advocate a Calibrate-and-Audit procedure, which uses a held-out audit split to estimate KPIs with Binomial inference, and show that sweeping calibration parameters yields constrained trade-offs and Pareto frontiers over operational behaviors. Empirical studies on simulations and molecular datasets illustrate how coverage-matched models can exhibit substantially different deployment profiles.

Overall, I am suggesting rejection, as the identified weaknesses (see below) do not appear to have straightforward or immediate remedies.

# Strengths

- The paper aims to address an important gap in bridging conformal prediction with decision making / the operational aspect of conformal prediction. There are multiple concurrent works in this area that I know of, and this paper definitely contributes to the discussion and is of interest to the ML community.

- The conceptual viewpoints (e.g., Pareto analysis, operational profiles, region-class tables) offered in this paper are interesting and have the potential to be extended to more general decision-making settings. The Pareto frontier shown and analyzed in Figure 5 is insightful.

# Weaknesses

- The presentation is bad, making the paper hard to follow. There are a couple of issues: (i) The paper uses terminologies (e.g., primitives, first-class, operational rates, semantic anchor) that are non-standard to the conformal prediction literature. This creates an unnecessary load for the audience to interpret these terms. Additionally, some terminologies are introduced without formal definitions, which may introduce ambiguity and hinder clarity in the presentation. (ii) The figures are not well-captioned. The existing caption is not very informative, and there are components in the figures left unexplained (e.g., in Fig. 2, what does the shaded area lying diagonally across the plot represent?). (iii) Many expressions in the paper are presented without verbal explanation, which also creates an unnecessary load for the audience to understand the equations.

- The paper seems to make a technical error. In Section 3.3, assuming exchangeability does not imply that $K_{\ell}^{\rm audit}(\theta)$ follows a binomial distribution. One would need a stronger i.i.d. assumption to prove this statement. Given that this is one of the fundamental statements underlying SSBC, the contribution of the paper would be significantly undermined if my observation is true.

- While the conceptual contribution is real, the paper lacks a true technical contribution. SSBC appears to be a relatively modest calibration adjustment of the conformal prediction method, which does not come with compelling advantages over the standard conformal prediction method. The core of Calibrate-then-Audit is essentially a standard data-splitting trick, which has already been widely used by many existing statistical frameworks.

**Additional Comments:**

None

**Audience:**

Yes

**Audience Explanation:**

This paper is of interest to the audience group of TMLR that is interested in decision-making under uncertainty, uncertainty quantification (conformal prediction), and operations research.

**Claims And Evidence:**

No

**Claims Explanation:**

The paper claims that the proposed small sample beta correction procedure achieves a PAC-style guarantee, but I did not find formal mathematical proof in the paper supporting this claim.

Additionally, the paper seems to be technically incorrect in one of its major contributions (see weakness).

**Requested Changes:**

- Please reduce the amount of nonstandard terminology used in the paper.

- Please add verbal explanations for important expressions that appear in the paper.

- Please refine the captions of the paper.

- Please correct the technical error (if my claim is true) in Section 3.3. Acknowledging the i.i.d. assumption would not be a valid fix for the problem, since the core of the conformal framework is its tolerance for exchangeable data. Therefore, the author might have to consider overhauling the proposed method.

- Could the authors also discuss the potential of the proposed framework extending to a continuous output space $\mathcal{Y}$? This is already mentioned in the final discussion of the paper, but I believe it deserves a more dedicated and detailed discussion.

---

> ### Author Response · Authors · 2026-04-27
>
> We thank the reviewer for the careful reading. The criticisms about presentation and definitions are well taken, and we respond point by point below.
> 1. Presentation and terminology. Agreed. The revision will reduce nonstandard terminology, define key quantities explicitly where they first appear, and rewrite the figure captions, including the shaded region, whose annotation was inadvertently removed during final editing.
> 2. Section 3.3. The reviewer is right: exchangeability alone does not give an exact Binomial law, and this will be corrected in revision. The intended setting is the post-calibration audit, where one studies rates of indicators induced by a fixed deployed rule. The revision will separate the audit layer by assumption: Hoeffding bounds as a conservative guardrail under independence and boundedness; empirical Bernstein as a variance-adaptive sharpening under the same conditions; and exact Bernoulli/Binomial intervals when the audit indicators are i.i.d. Bernoulli. The conformal validity layer remains separate from this.
> 3. Calibrate-and-Audit and the role of splitting. Sample splitting is of course classical, and we do not claim Calibrate-and-Audit as a novel splitting trick. The contribution is what the held-out split is used for: turning a fixed deployed conformal rule into an induced region-class summary from which multiple deployment KPIs can be read off the same object. Our contribution lies in this use of held-out auditing to characterize downstream properties, rather than in the splitting itself.
> 4. SSBC.The current draft does not contextualize SSBC relative to prior conformal PAC-style guarantees. Vovk (2012) establishes training-conditional $(\epsilon, \delta)$ validity for split conformal via Binomial-tail inversion. Marques (2025) subsequently showed that the training-conditional coverage is in fact exactly Beta-distributed, with parameters determined by the calibration size and the rank used for thresholding. We do not claim novelty for the Beta characterization itself. The contribution is operational: a coverage request should be specified as a pair $(\alpha, \delta)$, since a nominal level $\alpha$ alone leaves the strength of the guarantee unspecified. This matters in low-data regimes, where the calibration grid is coarse and several nearby grid indices are compatible with the same nominal target while delivering very different $\delta$. SSBC takes the user-specified $(\alpha^\star, \delta)$ as the primary object and deterministically selects the least conservative attainable grid index whose Beta-tail guarantee matches it, so that the deployed guarantee is the requested guarantee.
> 5. Formal support. Agreed. The PAC-style SSBC result will appear as a clearly labeled theorem, with its proof separated from the audit-layer discussion in (2).
> 6. Beyond the binary setting. This is underdeveloped in the current draft. The revision will expand the discussion and mark explicitly which parts of the framework are specific to binary split-conformal and which carry over.

---

> ### Comment · Reviewer_yfDs · 2026-04-29
>
> I thank the authors for their response. Since the authors acknowledged that the exact Binomial law statement is false, I recommend rejecting the paper and requiring a new round of revision. The statement is one of the paper's core contributions, and correcting it would require changes to other parts of the paper (e.g., the abstract and introduction).
>
> In addition, I expect fixing it to be a highly nontrivial task. What the authors seem to suggest in their rebuttal is to add an i.i.d. assumption to prove the statement using Hoeffding/Bernstein/Bernoulli arguments. However, i.i.d. is more restrictive than exchangeability, which fundamentally undermines its nature as a conformal prediction framework, and will make the proposed procedure less novel or even not novel. Therefore, I don't think adding the i.i.d. assumption is a satisfying solution to the problem. I would encourage the authors to explore further whether there exist feasible correction procedures under exchangeability alone.
>
> That said, while the paper shows potential, it contains fundamental technical flaws that cannot be adequately addressed within the scope of subsequent revisions. Therefore, I recommend rejection.

---

> > ### Author Response · Authors · 2026-04-30
> >
> > Dear Reviewer,
> >
> > We thank the reviewer for the thorough reading and careful engagement with the manuscript. We are grateful for the substantial time and thought reflected in the review, and we agree that the comments point to ways the paper can be improved substantially. In particular, we acknowledge that the manuscript requires a major revision to sharpen its presentation, clarify definitions and assumptions, and correct specific audit-layer claims. At the same time, we respectfully disagree with the recommendation for rejection. As the review itself notes, the paper addresses an important gap between conformal validity and deployment-facing decision making, and it offers a real conceptual contribution that may be of interest to the TMLR community. We believe that contribution remains intact: the paper makes visible the operational tradeoffs that practitioners need to understand when deploying conformal methods. We are thankful for the reviewer’s careful comments, and we believe they provide a constructive path toward a substantially stronger revision, through whatever mechanism the editor considers appropriate.

---

### Review · Reviewer_7Svu · 2026-03-31

**Summary Of Contributions:**

Contributions: This paper makes the argument that, when evaluating prediction sets, there are other metrics that matter besides coverage. For example, we may also care about how often the sets are empty, are of size 1, are of size 1 AND are correct, or do not contain a certain class. The paper makes the point that all of these metrics can be expressed as (linear) functions of the joint probability P(prediction set, true label). A method for estimating these metrics using a heldout dataset ("audit split") is suggested.

Strengths: The paper highlights the importance of evaluating conformal prediction sets in ways beyond coverage. This is a practically important direction for the field.

Weaknesses: The current presentation of ideas is unreadable, at both the micro (word/sentence level) and macro (organization) levels.

**Audience:**

No

**Audience Explanation:**

I can’t imagine a single human reader who would benefit from reading this paper.

**Claims And Evidence:**

No

**Claims Explanation:**

I don’t know who is the intended audience of this paper, but it doesn’t seem like it’s the conformal prediction community. It comes across as being written by AI, for AI. I found it to be almost unreadable. The jargon that is used is nonstandard but not defined (e.g., “semantics”, “KPIs”). Descriptions are vague, in the way that LLM outputs are when there is insufficient substance provided in the prompt. As a result, the underlying ideas of the paper, which are really quite simple, take a lot of effort to grasp. Significantly more thought needs to be put into the writing.

**Requested Changes:**

One concrete suggestion is to include a motivating example in the introduction to make the idea of KPIs more concrete. This could perhaps be the Tox21 example in the experiments section. It also feels like too much is deferred to the Appendix. The statement of theoretical guarantees for the proposed procedure should be included in the main text; the main text should read as a clean, complete story.

Beyond these suggestions, a lot more thought needs to be put into the writing overall.

---

> ### Author Response · Authors · 2026-04-27
>
> We thank the reviewer for the candid feedback. The criticisms about exposition and organization are well taken, and we respond point by point below.
> 1. Readability and presentation. Agreed. The intent was a deployment-oriented perspective on conformal sets, but the current draft uses too much nonstandard language and obscures the underlying ideas. The revision will simplify the writing, reduce jargon, define nonstandard terms explicitly where they first appear, and reorganize so that the main statistical objects and takeaways are introduced in a more concrete and conventional way.
> 2. Concreteness and motivating examples. The manuscript does attempt this: the introduction frames the problem in terms of deployment-facing quantities (commitment, deferral, decisive-error exposure), Figure 1 sketches the operational view in which calibration fixes a region partition and auditing produces a reusable summary, and Section 2.4 works through a binary example in which coverage is written as a sum of selected cells of the region–class table aiming to make explicit that coverage, singleton errors, and abstentions are projections of the same object. The reviewer's reaction shows this material is not prominent enough. The revision will pull the motivating example forward and lead with it.
> 3. "KPIs" and vague definitions. Fair. The current draft does define these quantities formally as Bernoulli indicators derived from the audited pair $(R_\tau(X), Y)$, with abstention and decisive error as examples, and shows that the region–class table is the reusable object from which such quantities are obtained by projection. But the definitions are not introduced plainly enough. We agree that a reader should not have to work this hard for a simple idea. The revision will use more standard language and make the "table first, metrics as projections" story explicit.
> 4. Organization and the appendix. The main paper currently leans too heavily on the appendix. The revision will bring the key theoretical guarantee statements into the main text, with the appendix supporting rather than carrying the argument.
> 5. The substance of the contribution. The reviewer characterizes the underlying idea as simple. Simplicity is not a weakness here: part of our contribution is to isolate a basic but under-emphasized point: once a prediction-set method is deployed, many practically relevant metrics are functions of the joint law of the prediction set and the true label, and these metrics are not determined by coverage alone. The audited region–class table and the projection identity are how we instantiate this. The work for revision is to make that core point transparent and to remove the presentation choices that currently obscure it.

---

### Review · Reviewer_voQV · 2026-04-07

**Summary Of Contributions:**

This paper conducts a decision-theoretic analysis of conformal prediction. Its stated contributions are as follows: First, it shows how many downstream utility functions can be written as (linear combinations of) probabilities of some event conditional on a prediction set and realized outcome. Second, it analyzes a finite-sample correction that can be used to generate probably-approximately-correct (PAC) guarantees for a conformal predictor, called the small-sample Beta correction (SSBC). In particular, the guarantees the SSBC provides are: with probability at least $1-\delta$ over the calibration set, the conformal set produced contains the true label with probability at least $1-\alpha$. Third, it analyzes how a hold-out sample can be used to estimate the downstream utilities associated with a conformal predictor and a fixed mapping from predictions to decisions (called Calibrate-then-Audit). Finally, in empirical experiments on several synthetic/real datasets, it validates coverage, hold-out evaluation, and optimal decisions implied by conformal predictions.

The main strength of this paper is its conceptual significance, particularly with respect to the first contribution above. It is useful that the paper formalizes how many quantities of interest about decisions using conformal predictions can be summarized by (convex combinations of) probabilities of some event conditional on the conformal prediction and realized outcome. For example, this formalism implies that computing estimated event probabilities on a held-out set can be sufficient for getting unbiased estimates of many different downstream utility functions. There is a very helpful “Worked example” at the end of section 2 explaining how to compute marginal coverage using these event probabilities. I wish the paper had spent more time in the body on conceptual insights into different commonly used utility functions, the assumptions about downstream tasks they reveal and related observations. Some of this is done in Appendix G.3.

I also found the discussion in “Binary conformal geometry and regime boundaries” to be interesting. In particular, the paper observes the following in the case where outcomes are binary: fixed thresholds used to generate conformal predictions can only produce some conformal sets but not others. In particular, depending on the thresholds, the conformal predictor may output both labels $\{0, 1\}$ or the empty set $\varnothing$ but cannot output both. This observation is implied directly from the fact that for binary outcomes $Y$, it holds $Pr(Y = 1 | A) + Pr(Y = 0 | A) = 1$ for any event $A$, but it provides nice intuition about the fact that the choice of thresholds impacts the set of possible outputs of the conformal predictor.

At best, this paper could provide a unified decision-theoretic analysis of the different metrics used to evaluate conformal predictors and the assumptions about downstream tasks underlying each choice of metric. Unfortunately, the paper instead spends significant space on the SSBC and Calibrate-then-Audit, for which I have significant reservations about the originality and technical significance.

The contribution associated with SSBC is a PAC-style conditional coverage statement. This work does not properly contextualize what the original contribution of SSBC is, if any, and there seem to be similar statements in, e.g., Vovk’s “Conditional validity of inductive conformal predictors” (https://arxiv.org/pdf/1209.2673), which also uses the regularized incomplete Beta function to get PAC-style guarantees. Moreover, the procedure called ‘Calibrate-then-Audit’ in the paper is a sample splitting procedure where a calibration sample is used to pick thresholds for (non)conformity scores and the audit sample is used to compute an estimate of the coverage guarantees, conditional on each prediction set. Neither the SSBC nor Calibrate-then-Audit procedures are technical contributions at all on their own. Both are straightforward applications of preexisting theory (PAC-style conformal guarantees in the first case, as far as I can tell, and the hold-out principle in the second case).

Similarly, the empirical experiments compare the proposed finite-sample correction with a version of conformal prediction with no finite-sample correction and the DKWM inequality. It compares these methods on marginal coverage. They also compare an evaluation using the hold-out sample Calibrate-then-Audit versus a leave-one-out sample and find the methods perform similarly. I believe the empirics serve to validate the theory in preceding sections and aren’t a significant contribution on their own.

Additionally, the exposition in the paper has significant flaws, with respect to terminology and the formalization of the results.

Terminology: There are often moments in the paper where terminology is used but never defined. I believe these are often intended to provide intuition or nuance, but instead I found them confusing and imprecise. For example, what is a “calibration navigation coordinate”? What is a “semantic anchor”? What is “scenario planning”? What are “operational profiles”, “operational rates”, “operational interface”, “operational view”, “operational evidence” etc.? Similarly for the following phrases: “calibration sweeps, Pareto views, leave-one-out proxies, and inverse-pricing analyses” and “simultaneous certification statements”. In all of these cases, the reader is left to try to decipher out meaning from context.

Formalization and descriptions of results:  For both Calibrate-and-Audit and SSBC, the definitions of procedures were defined only informally in the body of the paper. The definition of SSBC is the more egregious example of this. The manuscript states a PAC-style coverage guarantee but does not provide any description for how this guarantee is achieved, nor how to interpret it. The precise definition of SSBC is in appendix C.

**Additional Comments:**

There was no in-text description of Figure 1 whatsoever. Absent is misspelled in the third and fourth panels of Figure 2. Some parts of the figures are never explained. For example, it took me a while to figure out the darker blue lines / regions in Figure 2 are supposed to represent the support of a distribution.

**Audience:**

No

**Audience Explanation:**

The TMLR audience may be interested in the conceptual contributions I highlighted above (e.g., how common metrics are justified by different downstream tasks). I do not think the audience will appreciate the latter contributions of the paper, since originality and significance is limited. I believe changes that would fundamentally change the scope, clarity and contributions of the paper are necessary before some individuals would find the paper as a whole useful.

**Claims And Evidence:**

No

**Claims Explanation:**

I believe the claims in the paper are accurate, although their significance is limited. The strongest portions of the paper are the unifying decision-theoretic analysis of different commonly used performance metrics. I believe the second two contributions exist elsewhere in the literature, but there was no contextualization of how the proposed SSBC / Calibrate-then-Audit procedure is different from prior work or a novel contribution. The clarity of the paper was severely lacking.

**Requested Changes:**

- Contextualize the results, especially SSBC and calibrate-then-audit, within prior work.
- Define the SSBC in the body of the paper.
- Define the terms used in the paper; improve preciseness of language.

---

> ### Author Response · Authors · 2026-04-27
>
> We thank the reviewer for the careful reading. The criticisms about contextualization, framing, and definitions are well taken, and we respond point by point below.
> 1. The core contribution. We agree with the reviewer's characterization of the strongest part of the paper. The central conceptual contribution is the operational view: once a conformal rule is fixed, many downstream metrics are projections of the same audited region–class object, and coverage alone does not determine that object. Section 2.4 develops this in a worked example, writing coverage explicitly as a projection of the region–class table and indicating that other quantities follow analogously. The revision will lean harder into this contribution and spend less effort framing the later components as standalone methodological novelties.
> 2. SSBC and prior work. The current draft does not contextualize SSBC relative to prior conformal PAC-style guarantees. The relevant lineage is: Vovk (2012) establishes training-conditional $(\epsilon, \delta)$ validity for split conformal via Binomial-tail inversion, with Hoeffding as a looser variant. Marques (2025) subsequently showed that the training-conditional coverage of split conformal is in fact exactly Beta-distributed, with parameters determined by the calibration size and the rank used for thresholding. SSBC builds on this distributional result. We do not claim novelty for the Beta characterization itself, but instead claim an operational contribution. In small data settings, a coverage request should be specified as a pair $(\alpha, \delta)$, as a nominal level $\alpha$ alone does not determine a deployed rule, because it leaves the strength of the guarantee unspecified. For example, the same $\alpha$ paired with different $\delta$ defines operationally distinct requests: $\alpha = 0.05$ at $\delta = 0.01$ and $\alpha = 0.05$ at $\delta = 0.5$ are both nominally requests for 95% coverage, but the latter is essentially a coin-flip on whether the deployed rule meets the minimum requested target. SSBC takes the user-specified $(\alpha^\star, \delta)$ as the primary object and deterministically selects the least conservative attainable grid index whose Beta-tail guarantee matches it, so that the deployed guarantee is the least pessimistic version of the requested guarantee. The appendix gives this as a grid-index selection rule; the revision will surface it in the main text alongside explicit comparison to Vovk and Marques.
> 3. Calibrate-and-Audit and originality. Sample splitting is classical, and we do not claim Calibrate-and-Audit as a novel splitting principle. Our contribution lies in its use in this setting: calibration fixes a deployed conformal rule, the audit split estimates the induced region–class table, and a family of downstream quantities follows by projection. The contribution is therefore the use of held-out auditing to make a fixed conformal interface operationally legible, not hold-out evaluation per se. The main text already defines the region–class table and the projection identity linking it to event-functionals; the revision will elevate this as the central organizing idea and avoid overstating novelty at the level of ingredients.
> 4. Definitions in the main text. Agreed. The revision will move the formal definition of SSBC, including the selection rule, into the body, and will state Calibrate-and-Audit as a clear procedure rather than a narrative description. The current main text gives the guarantee and the role of the audit split, but the deterministic SSBC specification appears only in Appendix C, which is not sufficient.
> 5. Terminology and precision. Agreed. Terms such as "semantic anchor," "calibration navigation coordinate," "scenario planning," and several "operational ..." phrases were intended to build intuition but instead create interpretive burden. The revision will replace nonstandard terminology with standard statistical language and define any remaining special terms on first use. Figure 1 will get direct in-text discussion, and the captions will be made self-contained.
> 6. Role of the empirical section. The experiments are not intended as standalone empirical contributions. Their role is to support the theoretical development at the paper's scope: validating the SSBC, illustrating audit-based fixed-rule evaluation, and showing how planning views differ across coverage-matched rules. The revision will state this supporting role plainly and avoid the appearance of overclaiming.
> 7. Figures and presentation details. These are helpful concrete suggestions. The revision will add in-text discussion of Figure 1, correct the misspelling in Figure 2, and clarify in both caption and text that the darker blue region/lines in Figure 2 represent the support geometry.

---

### Decision · Action_Editor_o4zE · 2026-05-12

**Recommendation:** Reject

**Additional Comments:**

The paper is potentially interesting; a future submission should address the serious issues of the current paper. The incorrect Binomial claim and its assumptions should be addressed, and the conformal-validity and audit/inference claims should be clearly disentangled for classify. The exposition of the paper should be vastly improved; ensure that the use of AI does not deteriorate the clarity and directness of the writing. Main definitions and guarantees should be in the main body of the paper and related prior work should be sufficiently discussed.

**Audience:**

No

**Audience Explanation:**

No. While an operational perspective of conformal prediction may be relevant to some readers, the current manuscript is not sufficiently clear or technically sound - including incorrect results, hard to read exposition, and sufficiently missing related prior work.

**Claims And Evidence:**

No

**Claims Explanation:**

I concur with the reviewers that the claims are not supported by sufficiently accurate, convincing, and clear evidence. Reviewer yfDs identified a serious technical issue that the papers claimed Binomial law does not follow for exchangeability alone. Reviewers also found the conceptual contribution promising but found the technical claims around SSBC and Calibrate-and-Audit to be insufficiently contextualized relative to prior work and not clearly and not clearly established as novel or significant contributions. There was also significant concern about the exposition, undefined nonstandard terminology, and reliance on appendix material for the main contributions. Overall, while the paper raises an interesting deployment-oriented perspective, the submission does not provide sufficiently clear and technically sound evidence for its main claims.

**Resubmission Of Major Revision:**

The authors may consider submitting a major revision at a later time.